# PROOF-VERIFIER: ENABLING REINFORCEMENT LEARNING FROM VERIFIABLE REWARDS FOR MATHEMATICAL THEOREM PROVING

## ABSTRACT

Reinforcement Learning from Verifiable Rewards (RLVR) has revolutionized mathematical reasoning, enabling models like DeepSeek-R1 and OpenAI-o1 to achieve human-level performance on traditional math tasks where answers are single numbers or equations. However, extending RLVR to mathematical theorem proving remains challenging due to the fundamental verification bottleneck: unlike traditional math tasks, theorem proving generates entire reasoning processes that lack reliable automated verification methods for reward signal generation. In this work, we address this verification bottleneck by introducing PROOF-VERIFIER, the first generative verifier specifically designed to enable RLVR applications in mathematical theorem proving. PROOF-VERIFIER supports both formal and informal language (e.g., natural language) proofs, providing the detailed verification capabilities essential for effective reinforcement learning. To train PROOF-VERIFIER, we develop a formal-to-informal translation pipeline for high-quality synthetic data generation and employ a novel two-stage coarse-grained to fine-grained reward modeling mechanism. Experimental validation demonstrates that PROOF-VERIFIER achieves 93% verification accuracy, enabling reliable reward signals for RLVR applications. We show that PROOF-VERIFIER successfully enables effective test-time scaling (79% win rate in best-of-N sampling and 32% improvement in multi-turn proof refinement), and both single-turn and multi-turn RLVR training, consistently improving LLM-based theorem proving performance. Our work establishes the foundation for applying RLVR methodologies to mathematical theorem proving, extending the recent success of reasoning-enhanced models to this challenging domain.

## 1 INTRODUCTION

Recently, reasoning-enhanced LLMs such as DeepSeek-R1 (DeepSeek-AI et al., 2025) and OpenAI-o1 (OpenAI et al., 2024) have significantly reduced the performance gap between humans and artificial intelligence on traditional mathematical tasks (Lewkowycz et al., 2022) where the answer is a single number or equation. These methods employ Reinforcement Learning from Verifiable Rewards (RLVR) (Ouyang et al., 2022; Wang et al., 2025b), where reward signals are provided by comparing model outputs with reference answers, training models to generate the extended chain-of-thought reasoning (Wei et al., 2023) required to reach verifiable solutions on challenging benchmarks such as HMMT (HMMT, 2025), MATH-500 (Hendrycks et al., 2021; Lightman et al., 2024) and AIME (MAA, 2025).

Despite these advancements, extending RLVR to mathematical theorem proving remains challenging. Models that claim PhD-level competency continue to struggle with mathematical theorem proving problems at the high school or undergraduate level (Guo et al., 2025; Sheng et al., 2025). The primary bottleneck stems from the fundamental verification challenge: mathematical theorem proving tasks (Polu et al., 2022) require generating entire proof processes rather than single numbers or equations, making automated verification for reward signal generation significantly more complex. Moreover, diverse correct proofs can exist for the same statement, making comparisons with reference answers infeasible, which is a critical limitation for RLVR applications.

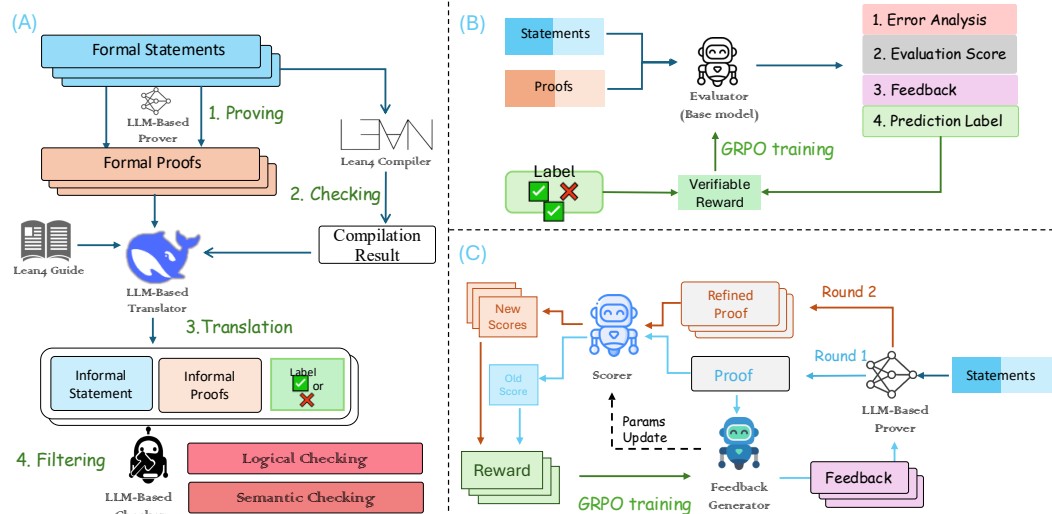

Figure 1: The training framework for PROOF-VERIFIER. (A) Formal-to-informal data synthesis pipeline with ATP verification and filtering. (B) Coarse-grained training with consistency-constrained label prediction. (C) Fine-grained training with proxy rewards from feedback-driven performance improvements.

While Automated Theorem Provers (ATPs) like Lean4 (de Moura et al., 2015; de Moura & Ullrich, 2021) can automatically verify proofs written in formal language by ensuring every deductive step conforms to a foundational logical system, they provide insufficient support for RLVR applications. First, they cannot handle informal language, yet LLMs perform better on natural language due to limited formal language representation in training data. Second, ATP verification results are coarse-grained binary labels that cannot distinguish between minor computational errors and fundamental logical flaws, both receiving the same "incorrect" label without guidance for targeted refinement.

To address this verification bottleneck and enable RLVR applications in mathematical theorem proving, we propose PROOF-VERIFIER, the first generative verifier for mathematical theorem proving tasks that supports both formal and informal language. PROOF-VERIFIER generates a comprehensive verification trajectory containing detailed error analysis, evaluation scores, actionable feedback, and final verification labels for each proof attempt, providing the reliable reward signals essential for effective reinforcement learning.

To support both formal and informal language, we develop a formal-to-informal translation pipeline with controlled generation and balanced labels (Figure 1 A). We sample formal proofs, verify them with ATPs, then translate to natural language with filtering strategies to ensure semantic consistency and logical correctness, achieving reliable data quality verified by human evaluation.

Based on this dataset, we train PROOF-VERIFIER using a novel two-stage coarse-to-fine-grained reward modeling mechanism (Figure 1 B, C). Since only coarse-grained labels are available initially, we design a progressive approach: (1) **Coarse-grained stage**: Label prediction with consistency constraints, where rewards require both accuracy and alignment with majority vote scores, enabling the model to learn robust structured reasoning processes. (2) **Fine-grained stage**: Proxy rewards are estimated by performance improvements brought by generated feedback (pink boxes in Figure 1), where feedback serves as refinement instructions to guide proof revision. This approach enables the model to develop fine-grained discriminative capabilities for distinguishing different error types and providing targeted refinement guidance.

Experimental validation demonstrates that PROOF-VERIFIER achieves 93% verification accuracy on our test set, establishing reliable reward signals for RLVR applications. We show that PROOF-VERIFIER successfully enables effective test-time scaling (Muennighoff et al., 2025), with superior response selection in best-of-N sampling settings (79% win rate) and multi-turn refinement instructions that improve LLM-based prover performance by 32%. Crucially, since PROOF-VERIFIER outputs both evaluation scores (usable as reward signals) and actionable feedback (serving as refine-

ment instructions), it naturally supports both single-turn and multi-turn RLVR training, achieving improved performance in both configurations and demonstrating the successful extension of RLVR methodologies to mathematical theorem proving. Finally, the detailed error analysis generated by PROOF-VERIFIER can be aggregated to help analyze and identify failure patterns for specific models, facilitating targeted improvements in model training iterations.

## 2 PROOF VERIFIER

### 2.1 DUAL-LANGUAGE DATASET CONSTRUCTION

Enabling RLVR for mathematical theorem proving requires training data that supports reliable reward signal generation across both formal and informal language proofs. Existing datasets present a critical gap: formal language datasets provide only binary ATP verification without fine-grained feedback, while informal language datasets lack reliable automated verification methods and may introduce validation errors, potentially overlooking logical issues and other subtle errors that are difficult to detect. To address this limitation, we construct a comprehensive dual-language dataset with controlled label quality and balanced coverage using our proposed formal-to-informal translation pipeline.

As shown in Figure 1, LLM-based provers generate 32 proof attempts for each formal statement, with the Kimina Lean Server (Santos et al., 2025) providing rigorous verification labels. DeepSeek-R1 then translates these verified formal statement-proof pairs into corresponding natural language versions. To enhance translation quality, we construct an llm.txt (Howard, 2024) file containing Lean4 syntax, tactics, and common proof methods as context manually. LLM-as-a-Judge is used to subsequently verify semantic consistency and logical correctness to ensure data quality after the conversion process. This formal-to-natural approach is more feasible than natural-to-formal translation, as understanding Lean4 syntax is simpler than generating it. The method ensures proof logic is rigorously compiler-verified before conversion, providing controlled generation of both correct and incorrect proofs with potential errors introduced only during the translation process.

To further enrich our training dataset distribution, we incorporate two additional natural language datasets: the OPC dataset (Dekoninck et al., 2025) provides labeled proof attempts on PutnamBench statements, while RFM Bench (Guo et al., 2025) contributes novel statements spanning different difficulty levels (high school to graduate) and mathematical domains (geometry, algebra, number theory, calculus). Processing details for both datasets are provided in Appendix H and M.

Our final training data comprises these three datasets with balanced sampling, maintaining a 1:1 ratio between natural language and formal language data. All three datasets contribute equally to the natural language data. The training datasets encompass a diverse range of mathematical domains, with detailed statistical distributions presented in Appendix F. For evaluation, we construct out-of-distribution test datasets using different statements and sampling models than those in training, ensuring our evaluation reflects generalization and robustness under distribution shift. The formal language test set contains 1,000 proof attempts with ATP-verified labels, while the natural language test set comprises 100 manually annotated statement-proof pairs with expert human verification.

### 2.1.1 QUALITY ASSESSMENT

Ensuring translation quality is critical for reliable reward signal generation in RLVR applications. While formal-to-informal translation introduces potential semantic gaps, we demonstrate that systematic filtering using LLM-as-a-Judge can effectively address these challenges. Analysis of 100 randomly sampled translation pairs reveals two primary error categories, which we successfully mitigate through targeted filtering strategies for semantic consistency and logical correctness:

**Statement Weakening**: Translations occasionally simplify formal statements to less restrictive conditions. We employ LLM-based filtering to detect and remove these cases while preserving valid proof-label pairs, since proofs for stronger claims remain valid for their weaker counterparts.

**Syntactic Copying**: Complex proofs sometimes result in direct code copying rather than natural language translation. We apply heuristic filtering rules that successfully eliminate these instances, achieving high precision on both validation and out-of-distribution evaluation sets. Detailed case analysis and filtering methodologies are provided in Appendix L.

## 2.2 TRAINING APPROACH

### 2.2.1 PROBLEM FORMULATION

We formalize the verification task of mathematical proofs as a structured generation task. Given a proof attempt $\tau$ and statement $x$, PROOF-VERIFIER generates a response sequence $r = (a, s, f, p)$ where $a \in \mathcal{A}$ represents error analysis, $s \in [0, 100]$ is the evaluation score, $f \in \mathcal{F}$ denotes feedback, and $p \in \{\text{True}, \text{False}\}$ is the binary correctness judgment. Let $\pi_\theta(r|\tau, x)$ denote our policy parameterized by $\theta$, and $y \in \{\text{True}, \text{False}\}$ be the ground truth label. We optimize $\theta$ to maximize expected reward $\mathbb{E}[R(r)]$ under different reward functions $R(\cdot)$ across two training stages, following a coarsed-grained to fine-grained training objective.

### 2.2.2 STAGE 1: CONSISTENCY-CONSTRAINED BINARY VERIFICATION

While RLVR training can achieve high accuracy on label prediction tasks, it often leads to high variance in intermediate reasoning chains. Even when models produce correct final judgments, their error analysis and evaluation scoring can be inconsistent across multiple evaluations of the same proof. This inconsistency poses two critical problems: (1) it undermines the model's reliability for fine-grained evaluation tasks where consistent scoring is essential, and (2) it creates unstable training dynamics for Stage 2, which depends on reliable score distributions as reward signals.

To address this challenge, we introduce consistency constraints that enforce both accuracy verification results and consistent evaluation scores. For each input $(\tau, x)$, we generate $N$ parallel samples $\{r_1, r_2, \ldots, r_N\}$ and define the correct prediction set as $\mathcal{C} = \{r_i : p_i = y\}$ and score mode: $s_{\text{mode}} = \arg\max_s |\{r_i \in \mathcal{C} : s_i = s\}|$.

Our reward function enforces both accuracy and consistency:

$$R_1(r_i) = 2 \cdot \mathbb{I}[p_i = y \text{ and } s_i = s_{\text{mode}}] - 1 \tag{1}$$

The intuition behind this design leverages the autoregressive generation order where error analysis and scoring precede the final prediction label. By enforcing consistency in the intermediate steps while supervising only the final binary judgment, the model learns to develop stable, coherent reasoning processes that support accurate predictions, which is crucial for the fine-grained capabilities developed in Stage 2. Detailed theoretical analysis demonstrating the convergence properties of this consistency-constrained approach is provided in Section B.1.

### 2.2.3 STAGE 2: FEEDBACK QUALITY OPTIMIZATION

Stage 1 enables the model to distinguish correct from incorrect proofs but lacks fine-grained discriminative power to assess varying degrees of proof quality. Stage 2 addresses this limitation by leveraging our sequential generation order where evaluation scores precede feedback. This temporal structure enables mutual supervision: fine-grained error analysis leads to more precise scores, which in turn enables more effective feedback generation.

We initialize both feedback provider $F_\theta$ and scorer $S_\phi$ with Stage 1 parameters: $\theta^{(0)} = \phi^{(0)} = \theta_{\text{Stage1}}$. The training process operates through a multi-step feedback refinement loop. Given a mathematical statement $x$, an external prover $P$ first generates an initial proof attempt $\tau_0$, which the scorer $S_\phi$ evaluates to produce a baseline score $s_0 = S_\phi(\tau_0, x)$. The feedback provider $F_\theta$ then generates $n$ diverse feedback responses $\{f_1, f_2, \ldots, f_n\}$ based on the initial proof and statement. Each feedback $f_i$ is provided to the prover $P$, which attempts to incorporate the suggestions and produce a revised proof $\tau_{1,i} = P(\tau_0, f_i, x)$. The scorer evaluates these revised proofs, yielding new scores $s_{1,i} = S_\phi(\tau_{1,i}, x)$ for each feedback-guided revision.

The key insight is that better feedback should lead to improved proofs, as measured by score increases. We therefore define the reward for feedback $f_i$ based on the score improvement it enables:

$$R_2(f_i) = \text{sign}(s_{1,i} - s_0 - \delta) \cdot \mathbb{I}[|s_{1,i} - s_0| > \delta] \tag{2}$$

where $\delta = 10$ filters out minor score fluctuations to focus on meaningful improvements.

To address the instability inherent in jointly optimizing both the feedback provider $F_\theta$ and scorer $S_\phi$, we employ a momentum encoder strategy (He et al., 2020) with differentiated update frequencies.

The feedback provider parameters are updated at every training step, while the scorer parameters remain frozen for $m = 100$ steps before being updated to match the current feedback provider parameters: $\phi^{(t+1)} \leftarrow \theta^{(t)}$ when $t \bmod m = 0$, and $\phi^{(t+1)} = \phi^{(t)}$ otherwise. Theoretical analysis for this design and the effects of momentum encoder updating strategy for training robustness is provided in Section B.2.

These two stages all use the standard GRPO algorithm for parameters updating:

$$\mathcal{J}_{GRPO}(\theta) = \mathbb{E}\left[\min\left[\frac{\pi_\theta(o|q)}{\pi_{\theta_{old}}(o|q)}\hat{A}, \text{clip}\left(\frac{\pi_\theta(o|q)}{\pi_{\theta_{old}}(o|q)}, 1-\epsilon, 1+\epsilon\right)\hat{A}\right] - \beta\mathbb{D}_{KL}[\pi_\theta||\pi_{ref}]\right] \quad (3)$$

The complete training procedure is summarized in Algorithm 1. Detailed justification and experimental results for using GRPO exclusively without supervised fine-tuning is provided in Appendix C.

---

**Algorithm 1** Two-Stage PROOF-VERIFIER Training

---

**Require:** Dataset $\mathcal{D} = \{(\tau_i, x_i, y_i)\}$, prover model $P$, momentum interval $m = 100$
1: Initialize $\theta^{(0)}$ with Qwen3-8B parameters
2: **Stage 1: Consistency-Constrained Training**
3: **for** $t = 1$ to $T_1$ **do**
4:     Sample batch $\{(\tau_k, x_k, y_k)\}_{k=1}^B \sim \mathcal{D}$
5:     **for** each $(\tau_k, x_k, y_k)$ in batch **do**
6:         Generate $N$ responses: $\{r_{k,i} = (a_{k,i}, s_{k,i}, f_{k,i}, p_{k,i})\}_{i=1}^N \sim \pi_\theta(\cdot|\tau_k, x_k)$
7:         Define correct prediction set: $\mathcal{C}_k = \{r_{k,i} : p_{k,i} = y_k\}$
8:         Compute score mode: $s_{\text{mode},k} = \arg\max_s |\{r_{k,i} \in \mathcal{C}_k : s_{k,i} = s\}|$
9:         Compute rewards: $R_1(r_{k,i})$ using Equation 1
10:    **end for**
11:    Update via GRPO: $\theta \leftarrow \theta + \alpha\nabla_\theta\mathcal{J}_{GRPO}(\theta)$ using Equation 3
12: **end for**
13: Initialize scorer: $\phi^{(0)} \leftarrow \theta^{(T_1)}$
14: **Stage 2: Feedback Quality Optimization**
15: **for** $t = 1$ to $T_2$ **do**
16:    Sample mathematical statements: $\{x_j\}_{j=1}^M$
17:    **for** each statement $x_j$ **do**
18:        $\tau_{0,j} \leftarrow P(x_j)$ {Generate initial proof}
19:        $s_{0,j} \leftarrow S_\phi(\tau_{0,j}, x_j)$ {Score initial proof}
20:        Generate $n$ feedback: $\{f_{i,j}\}_{i=1}^n \sim F_\theta(\cdot|\tau_{0,j}, x_j)$
21:        **for** each feedback $f_{i,j}$ **do**
22:            $\tau_{1,i,j} \leftarrow P(\tau_{0,j}, f_{i,j}, x_j)$ {Revise proof with feedback}
23:            $s_{1,i,j} \leftarrow S_\phi(\tau_{1,i,j}, x_j)$ {Score revised proof}
24:        **end for**
25:        Compute rewards: $R_2(f_{i,j})$ using Equation 2
26:    **end for**
27:    Update via GRPO: $\theta \leftarrow \theta + \alpha\nabla_\theta\mathcal{J}_{GRPO}(\theta)$ using Equation 3
28:    **if** $t \bmod m = 0$ **then**
29:        Momentum update: $\phi \leftarrow \theta$ {Transfer knowledge to scorer}
30:    **end if**
31: **end for**
32: **return** $\theta^{(T_2)}$

---

## 3 EFFECTIVENESS OF THE PROOF-VERIFIER

In all subsequent experiments, PROOF-VERIFIER uses simple prompts with parallel sampling, as our comparative experiments demonstrates this configuration provides optimal robustness and consistency for evaluation and verification, as detailed in Appendix C.1.

### 3.1 PERFORMANCE ANALYSIS

To verify the reliability of PROOF-VERIFIER for RLVR applications, we first evaluate its verification accuracy and the correlation of fine-grained scores for model ranking tasks.

Table 1: Performance comparison of different language models on the mathematical proof evaluation task.

| Verifier | Natural Language | | | | Formal Language | | | |
|---|---|---|---|---|---|---|---|---|
| | Acc | Prec | Rec | F1 | Acc | Prec | Rec | F1 |
| Qwen3-8B | 0.57 | 0.48 | 0.63 | 0.54 | 0.62 | 0.52 | 0.68 | 0.59 |
| Qwen2.5-72B | 0.53 | 0.44 | 0.58 | 0.50 | 0.58 | 0.48 | 0.63 | 0.54 |
| Magistral | 0.59 | 0.50 | 0.65 | 0.57 | 0.64 | 0.54 | 0.70 | 0.61 |
| Gemma | 0.58 | 0.49 | 0.64 | 0.56 | 0.63 | 0.53 | 0.69 | 0.60 |
| Qwen3-235B | 0.71 | 0.62 | 0.75 | 0.68 | 0.76 | 0.67 | 0.80 | 0.73 |
| Deepseek-R1 | 0.73 | 0.64 | 0.76 | 0.69 | 0.78 | 0.69 | 0.81 | 0.75 |
| **Ours** | **0.93** | **0.93** | **0.94** | **0.93** | **0.91** | **0.90** | **0.91** | **0.90** |

**Verification Accuracy**: Table 1 Left presents the verification accuracy results across both language modalities. For natural language proofs, PROOF-VERIFIER significantly outperforms larger open-source models on metrics including accuracy and F1, achieving better correlation with human evaluation, which demonstrates reliable scoring and more consistent alignment with human preferences. The primary error source in existing open-source models is their tendency to incorrectly classify false proofs as correct, exhibiting high recall but low precision, indicating insufficient error detection capabilities. For formal language verification (Table 1 Right), PROOF-VERIFIER achieves superior accuracy compared to other models without access to compilation results, demonstrating the model's ability to better interpret and evaluate formal language proofs. When compilation results are included, PROOF-VERIFIER achieves 0.98 accuracy, with ATP verification serving as the lower bound. Despite this high baseline, PROOF-VERIFIER maintains significant advantages through its generated feedback, which provides substantially greater utility than ATP compilation results. Comparative examples illustrating this advantage are provided in Figure 6 in Appendix.

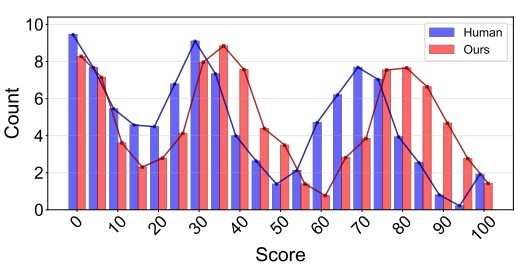

Figure 2: Distribution comparison between human and model evaluation scores using 100 randomly sampled items from the test dataset.

**Ranking Correlation**: We further validate the reliability of fine-grained evaluation scores by comparing model rankings derived from these scores against ground truth rankings. For natural language proofs, ground truth rankings are derived from human evaluation scores, while formal language rankings are computed using Pass@32 metrics. The Pearson correlations are 0.83 and 0.91 respectively at the individual item level, while completely consistent at the model level. The score distributions of our model and human evaluation are shown in Figure 2, demonstrating the effectiveness of our training method. While PROOF-VERIFIER generates consistently higher scores than human evaluators, the distributions maintain the same trend. This systematic offset reflects LLM-as-a-judge's higher tolerance for errors due to weaker error detection capabilities, but the relative quality rankings remain accurate as shown by strong correlations.

## 3.2 ABLATION STUDIES

As shown in Table 2, incorporating our synthetic formal-to-informal translated data improves verification performance on both language modalities, demonstrating that controlled label quality and increased data diversity are crucial for reliable reward generation. For the training methodology, the consistency constraint significantly reduces score variance across multiple samples of the same proof-statement pair, providing the stable evaluation capabilities necessary for reliable reference standards in subsequent training stages. Building on this stability foundation, the fine-grained feedback training component increases discriminative power across different proof qualities, as evidenced by improved score distributions that better differentiate between varying proof attempts. To further verify the theoretical feasibility and effectiveness of our proposed Stage 2 method, we employ human evaluators to conduct pairwise comparisons of verification trajectories, where we separately rank the error analysis, actionable feedback, and refined proof attempts from each trajectory.

The correlation between error analysis and feedback rankings is 0.90, while the correlation between feedback and refined proof attempt rankings is 0.95. This high consistency validates our hypothesis that better error analysis leads to better feedback generation, which in turn produces better proof refinements. Details are provided in Appendix D.

Table 2: Ablation study results.

| Setting | ACC | ICC | Variance |
|---|---|---|---|
| *Data Ablation* | | | |
| OPC | 0.82 | 0.53 | 0.13 |
| + RFM | 0.87 | 0.52 | 0.11 |
| + Ours | 0.91 | 0.55 | 0.14 |
| *Method Ablation* | | | |
| Baseline | 0.91 | 0.55 | 0.14 |
| + Consistency Constraint | 0.91 | 0.57 | **0.09** |
| + Proxy Reward | **0.93** | **0.57** | 0.11 |

### 3.3 CASE ANALYSIS

We evaluated the quality of generated error analysis and actionable feedback using human annotators across both formal and natural language. Human evaluation shows that PROOF-VERIFIER locates approximately 81% of errors when averaged among all annotators, indicating the model learns detailed analysis of the entire proof process to identify errors in each proving step. For feedback, annotators considered 87% of the generated feedback to be useful and actionable, capable of helping prover models correct errors in previous attempts. This demonstrates that PROOF-VERIFIER generates the high-quality feedback essential for effective RLVR applications and suggests potential for performance improvement during inference, which is discussed in Section 4.1. Case examples are provided in Appendix N.

## 4 ENABLING RLVR FOR MATHEMATICAL THEOREM PROVING

To validate that PROOF-VERIFIER provides reliable reward signals for RLVR applications, we first demonstrate its effectiveness in test-time scaling scenarios, which represent a preliminary but essential validation of our verifier's practical utility for reinforcement learning settings.

### 4.1 BEST-OF-N

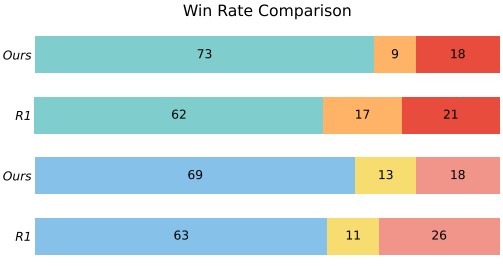

Figure 3: Win rate comparison between models on Natural Language (top two rows) and Formal Language (bottom two rows) tasks. The segments from left to right represent win, tie, and loss rates respectively. R1 means Deepseek-R1-0528.

We employ PROOF-VERIFIER evaluation scores to rank proof attempts from LLM-based provers in best-of-N sampling settings, selecting the highest-scoring attempt. For natural language evaluation, we compare the selected proof against single-sample baselines through human preference evaluation, measuring win rates where best-of-N selections outperform or tie with single samples. For formal language evaluation, we use ATP verification to determine success, with ties occurring when both best-of-N and single-sample approaches pass verification. Experimental results (Figure 3) demonstrate that our model's selected samples achieve higher win rates compared to single samples and outperform the DeepSeek-R1 model on our test dataset. This preliminary validation confirms that PROOF-VERIFIER generates evaluation signals that effectively guide proof selection, establishing the foundation for more comprehensive RLVR applications in subsequent experiments.

### 4.2 REFINEMENT BASED ON FEEDBACK

To validate that PROOF-VERIFIER feedback enables iterative improvement essential for multi-turn RLVR applications, we evaluate refinement capabilities across both language modalities. For natural language proofs, we assess whether refined proofs show improvement through human evaluation. For Lean4, we use ATP verification to compare pass@k performance before and after refinement. **Natural Language:** Human annotators found that 73% of the feedback effectively identifies errors and provides actionable guidance. However, only 51% of errors were successfully addressed in the refinement process, while 17% of proofs showed minimal changes and 32% introduced new errors.

This gap between feedback effectiveness (73%) and successful error correction (51%) reveals limitations in current LLM-based provers' ability to utilize external guidance, highlighting an important direction for future RLVR training that focuses on improving feedback utilization capabilities. **Formal Language:** For Lean4 formal proofs, refinement improved pass@k performance from 37% to 51%, demonstrating that PROOF-VERIFIER feedback provides valuable guidance for correcting formal proof errors. The feedback primarily addresses: 1) correcting boundary condition assumption errors, 2) clarifying unclear theorem scope, and 3) resolving incomplete proofs that use `sorry` placeholders to skip proof goals. This 14-point improvement demonstrates that feedback-guided refinement achieves better scaling results with fewer sampling attempts compared to sequential and parallel scaling approaches, validating the practical utility of our verifier for RLVR applications.

## 4.3 REINFORCEMENT LEARNING

### 4.3.1 SINGLE-TURN & MULTI-TURN RL

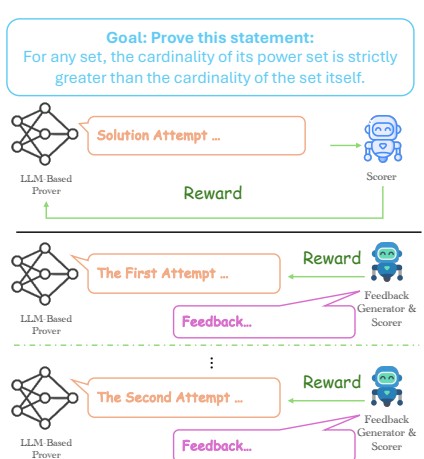

Figure 4: Comparison between single-turn and multi-turn reinforcement learning approaches.

Having demonstrated that PROOF-VERIFIER provides reliable evaluation scores and actionable feedback for test-time applications, we now validate the core claim of our work: that our verifier enables effective RLVR training for mathematical theorem proving. This represents the ultimate test of whether we have successfully addressed the verification bottleneck that previously prevented RLVR applications in this domain. We evaluate PROOF-VERIFIER in reinforcement learning settings under two configurations that reflect different RLVR paradigms (as shown in Figure 4). For training, we use Qwen3-8B for informal language proofs and DeepSeek Prover 2 for formal language proofs. In single-turn RL, PROOF-VERIFIER acts as a reward model, providing binary reward signals based on label predictions (True/False) for each proof attempt, which validates the basic RLVR functionality where our verifier provides the verifiable reward signals that were previously missing. In multi-turn RL, PROOF-VERIFIER serves dual roles: as a feedback provider that engages in multi-turn conversations by providing refinement instructions, and as a reward model that supplies binary reward signals based on correctness predictions for each round of proof attempts. This configuration demonstrates the full potential of our approach, enabling iterative proof development guided by detailed feedback.

### 4.3.2 EXPERIMENTAL RESULTS

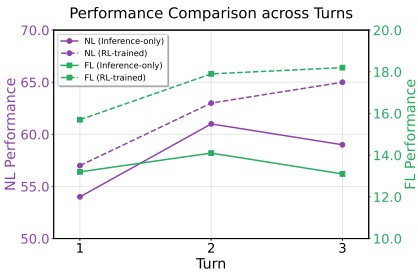

Figure 5: Multi-turn refinement performance comparison.

Our experimental results demonstrate that RL training enables the model to significantly reduce errors and improve output quality compared to the baseline model (Figure 5). Notably, the quality of individual sample generation improved substantially, narrowing the performance gap with best-of-n sampling and achieving higher win rates against reference solutions.

In our multi-turn RL framework, the model receives two complementary signals at each turn: (1) explicit feedback from the verifier, provided as in-context information to guide the prover model, and (2) scalar reward scores from the verifier, quantifying the relative quality of the current turn's proof attempt. The prover model's observation at turn $t$ consists of the complete interaction history $\{p_0, f_0, p_1, f_1, \ldots, p_{t-1}, f_{t-1}\}$, while the verifier focuses solely on the current proof attempt $p_t$ for scoring and feedback generation, where $p_i$ and $f_i$ represent the proof and feedback at turn $i$, respectively. Crucially, reward signals are attributed only to the current turn's actions, thereby simplifying the credit assignment problem.

Multi-turn RL training yields two improvements: First, the model's capacity to utilize feedback improves, with the gap between feedback quality and actual proof improvement narrowing compared to prompt-based refinement approaches. Second, the performance degradation commonly observed in multi-turn feedback-based systems is mitigated, as the prover becomes more robust in leveraging beneficial feedback while avoiding deterioration from erroneous guidance (as shown in Figure 5). This improvement can be explained by our designed reward structure and credit assignment strategy. By optimizing only the current turn's proof generation regardless of feedback correctness, the model learns to selectively utilize accurate feedback for enhanced proof quality while developing resilience against noisy or incorrect feedback. This approach implicitly trains the model to maintain correct proofs and refine incorrect ones across diverse scenarios, leading to improved robustness in multi-turn interactions.

## 5 RELATED WORK

**Mathematical theorem proving** has seen significant progress with benchmarks like miniF2F (Zheng et al., 2022), FIMO (Liu et al., 2023), and PutnamBench (Tsoukalas et al., 2024), alongside advances in AI-assisted approaches such as AlphaGeometry (Trinh et al., 2024; Chervonyi et al., 2025) and recent neural theorem provers (Polu & Sutskever, 2020). Current verification relies primarily on interactive theorem provers like Lean4 (de Moura et al., 2015; de Moura & Ullrich, 2021), Coq (Huet & Paulin-Mohring, 2000), and Isabelle (Nipkow et al., 2002), with recent improvements in compilation efficiency through Kimina Lean Server (Santos et al., 2025). However, these approaches have critical limitations for RLVR applications: they only support formal languages while excluding natural language proofs, and provide only binary verification results without the detailed error analysis or actionable feedback essential for effective reinforcement learning.

**Reward modeling for mathematical reasoning** has primarily focused on traditional problem-solving tasks. Process reward models (Lightman et al., 2024; Wang et al., 2024a) have shown success in step-by-step verification for computational problems, while outcome reward models achieve strong results through final answer comparison (Cobbe et al., 2021). More broadly, LLM-as-a-judge approaches (Zheng et al., 2023) have evolved from instruction-following evaluation to reward models across domains including medical QA (Croxford et al., 2025), multimodal tasks (Chen et al., 2024), and code generation (Zhao et al., 2024), with open-source alternatives like PandaLM (Wang et al., 2024b) and Prometheus (Kim et al., 2024a;b) providing cost-effective solutions. While these methods succeed in traditional mathematical tasks through reference answer comparison (Chen et al., 2025), theorem proving's process-oriented nature and diverse solution paths present unique verification challenges that limit RLVR applications in this domain. Our work addresses this verification bottleneck by developing the first dual-language verifier specifically designed for mathematical theorem proving, enabling RLVR applications through detailed error analysis and fine-grained evaluation capabilities via novel two-stage reward modeling.

## 6 CONCLUSION

We introduce PROOF-VERIFIER, a generative verifier that enables effective verification for mathematical theorem proofs in both formal and natural languages. Through a novel data synthesis pipeline and two-stage coarse-to-fine training framework, our model learns to provide quantitative assessment and detailed qualitative feedback for proof attempts. Experimental results show that PROOF-VERIFIER achieves strong verification accuracy with high correlation to human judgment and ATP in model ranking. Our verifier effectively supports test-time scaling through best-of-n selection and enables iterative proof refinement via actionable feedback. Finally, we demonstrate that PROOF-VERIFIER serves as an effective verifiable reward model for RLVR, revealing the potential of LLM-based verifiers to advance automated reasoning in domains without reference answers.

## ETHICS STATEMENT

This work proposes PROOF-VERIFIER to enable Reinforcement Learning from Verifiable Rewards for mathematical theorem proving, enhancing models' mathematical reasoning capabilities. By

generating detailed mathematical proof processes rather than potentially error-prone intermediate steps, our approach increases the trustworthiness of LLM outputs. This advancement strengthens the reliability and educational value of AI systems for mathematical learning and instruction, promoting more transparent and verifiable mathematical reasoning.

## REPRODICIBILITY STATEMENT

To ensure reproducibility of our results, we provide comprehensive implementation details and experimental specifications throughout the paper and supplementary materials. Section 2 and Section 4 detail our training methodology, model architectures, and experimental setup, while the complete data processing pipeline, including our formal-to-informal translation procedure and filtering strategies, is described in Section 2.1 and Appendix F. All model configurations, hyperparameters, training procedures, and prompt templates used in our experiments are documented in Appendices G and M. To validate the quality of our human evaluation process, we include detailed examples of our annotation interface and inter-annotator agreement scores in Appendix D, along with sample cases demonstrating the assessment criteria used by human evaluators. The statistical distributions of our training datasets, experimental configurations for all baseline comparisons, and additional implementation details are provided in Appendices F through N. All materials necessary for reproducing our experimental results, including dataset construction procedures, evaluation protocols, and analysis methods, are included in the submitted appendices.

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

## A  USE OF LLMs

We used Grammarly's AI assistance to assist with grammar checking and sentence-level polishing.

```
=== Lean4 Code lean4_code_1 ===
<ERROR_SUMMARY>
Total errors found: 1
</ERROR_SUMMARY>

<ERROR #1>
<ERROR_LOCATION>
Line 11, columns 2-5
</ERROR_LOCATION>

<ERROR_TYPE>
Single-line error
</ERROR_TYPE>

<ERROR_TEXT>
"rw"
</ERROR_TEXT>

<SURROUNDING_CONTEXT>
     9:      ext f
    10:      simp [h_sols]
>>> 11:    rw [h₁]
    12:    rfl
</SURROUNDING_CONTEXT>

<ERROR_MESSAGE>
maximum recursion depth has been reached
use `set_option maxRecDepth <num>` to increase limit
use `set_option diagnostics true` to get diagnostic information
</ERROR_MESSAGE>
</ERROR #1>
```

The error occurs because of incorrect `rw` tactic syntax on line 11. The immediate issue is ** `rw [h:]` should be `rw [h]` ** - remove the colon as it's invalid syntax.

If fixing the syntax doesn't resolve the recursion error, the problem is likely that hypothesis `h` creates an infinite rewrite loop. This happens when `h` is an equality that rewrites a term back to itself. Here are the solutions to try in order:

1. **Fix syntax**: Change `rw [h:]` to `rw [h]`
2. **Try reverse direction**: Use `rw [← h]` instead
3. **Use alternative tactics**: Try `simp only [h]`, `exact h`, or `apply h` depending on what `h` proves
4. **Debug the hypothesis**: Add `#check h` before line 11 to verify `h` exists and see its type
5. **Enable tracing**: Use `set_option trace.Meta.Tactic.rewrite true` before the rewrite to see what's happening

The recursion depth error typically indicates that the rewrite rule is being applied infinitely, so switching tactics or rewrite direction usually resolves it.

(a) Lean4 Compilation Results.  (b) Feedback generated by PROOF-VERIFIER.

Figure 6: Error analysis and feedback generation comparison.

## B  THEORETICAL ANALYSIS

### B.1  CONSISTENCY-CONSTRAINED TRAINING

The consistency constraint in our Stage 1 reward function is motivated by a fundamental principle in mathematical evaluation: independent correct assessments of the same proof should converge to similar conclusions. To formalize this intuition, we begin by establishing the mathematical framework.

Given $N$ parallel samples $\{r_1, r_2, \ldots, r_N\}$ for a proof attempt, each response $r_i$ produces a binary prediction $p_i \in \{\text{True}, \text{False}\}$ and evaluation score $s_i \in [0, 100]$. Let $y \in \{\text{True}, \text{False}\}$ denote the ground truth label. We model the policy as $\pi_\theta(r_i|x)$ where $x$ represents the input proof attempt.

Our fundamental assumption is that for a well-trained evaluator, the score distribution conditioned on correct predictions should concentrate around some true value. Formally, if we denote by $\mathcal{C} = \{r_i : p_i = y\}$ the set of correct predictions, then as the model quality improves, we expect:

$$s_i | p_i = y \xrightarrow{d} \delta_{\mu^*} \tag{4}$$

where $\delta_{\mu^*}$ is a point mass at the true evaluation score $\mu^*$, and $\xrightarrow{d}$ denotes convergence in distribution.

However, during training, we observe significant variance in scores even among correct predictions. To address this, we employ a consistency constraint based on modal consensus. Consider the empirical distribution of scores among correct predictions:

$$P_{\text{emp}}(s) = \frac{1}{|\mathcal{C}|} \sum_{r_i \in \mathcal{C}} \mathbf{1}[s_i = s] \tag{5}$$

For any estimator $\hat{s}$ of the consensus score, we can define the 0-1 consensus risk as:

$$R(\hat{s}) = \mathbb{E}[\mathbf{1}[s \neq \hat{s}]|p = y] = \sum_{s'} P(s = s'|p = y)\mathbf{1}[s' \neq \hat{s}] \tag{6}$$

Expanding this expression:

$$R(\hat{s}) = \sum_{s' \neq \hat{s}} P(s = s'|p = y) \tag{7}$$

$$= 1 - P(s = \hat{s}|p = y) \tag{8}$$

The mode estimator $\hat{s}_{\text{mode}} = \arg\max_s P(s|p = y)$ minimizes this risk since:

$$\hat{s}_{\text{mode}} = \arg\max_s P(s|p = y) = \arg\min_s [1 - P(s|p = y)] = \arg\min_s R(\hat{s}) \tag{9}$$

Furthermore, the mode exhibits superior robustness properties. Under $\epsilon$-contamination where a fraction $\epsilon$ of the correct predictions are replaced by adversarial scores, the empirical distribution becomes:

$$P_{\text{cont}}(s) = (1 - \epsilon)P_{\text{emp}}(s) + \epsilon P_{\text{adv}}(s) \tag{10}$$

where $P_{\text{adv}}(s)$ is the adversarial distribution. The mode remains stable as long as $\epsilon < \frac{1}{2} - \frac{1}{2|\mathcal{S}|}$ where $|\mathcal{S}|$ is the number of distinct score values, while the sample mean can be arbitrarily shifted by any $\epsilon > 0$.

Now we analyze our reward function. The consistency-constrained reward can be written as:

$$R_1(r_i) = \mathbf{1}[p_i = y] \cdot (2\mathbf{1}[s_i = s_{\text{mode}}] - 1) - \mathbf{1}[p_i \neq y] \tag{11}$$

To understand the expected behavior, we compute the expected reward. Let $A = \{p_i = y\}$ denote the accuracy event and $C = \{s_i = s_{\text{mode}}\}$ denote the consistency event. Then:

$$\mathbb{E}[R_1(r_i)] = \mathbb{E}[\mathbf{1}[A] \cdot (2\mathbf{1}[C] - 1)] - \mathbb{E}[\mathbf{1}[A^c]] \tag{12}$$

$$= \mathbb{E}[\mathbf{1}[A] \cdot 2\mathbf{1}[C]] - \mathbb{E}[\mathbf{1}[A]] - P(A^c) \tag{13}$$

$$= 2\mathbb{E}[\mathbf{1}[A \cap C]] - P(A) - P(A^c) \tag{14}$$

$$= 2P(A \cap C) - P(A) - (1 - P(A)) \tag{15}$$

$$= 2P(A \cap C) - 1 \tag{16}$$

Using the conditional probability identity $P(A \cap C) = P(C|A)P(A)$:

$$\mathbb{E}[R_1(r_i)] = 2P(C|A)P(A) - 1 \tag{17}$$

$$= P(A)[2P(C|A) - \frac{1}{P(A)}] \tag{18}$$

$$= P(A)[2P(s_i = s_{\text{mode}}|p_i = y) - \frac{1}{P(A)}] \tag{19}$$

For the expected reward to be positive, we need:

$$P(s_i = s_{\text{mode}}|p_i = y) > \frac{1}{2P(A)} \tag{20}$$

This inequality reveals a crucial trade-off: when accuracy $P(A)$ is low, the consistency requirement becomes more stringent. However, as accuracy improves ($P(A) \to 1$), the consistency threshold approaches $\frac{1}{2}$, making positive rewards more achievable.

To analyze the training dynamics, consider the policy gradient:

$$\nabla_\theta \mathbb{E}[R_1(r_i)] = \mathbb{E}[R_1(r_i)\nabla_\theta \log \pi_\theta(r_i)] \tag{21}$$

Expanding using our reward decomposition:

$$\nabla_\theta \mathbb{E}[R_1(r_i)] = \mathbb{E}[\mathbf{1}[p_i = y](2\mathbf{1}[s_i = s_{\text{mode}}] - 1)\nabla_\theta \log \pi_\theta(r_i)] \tag{22}$$
$$- \mathbb{E}[\mathbf{1}[p_i \neq y]\nabla_\theta \log \pi_\theta(r_i)] \tag{23}$$

This gradient has two components. The first term encourages both accuracy and consistency simultaneously, while the second term discourages incorrect predictions. The key insight is that the gradient magnitude for consistency is proportional to the accuracy level, creating a self-reinforcing dynamic.

As training progresses, we expect the accuracy $P(p_i = y)$ to increase. When $P(p_i = y) \to 1$, the expected reward simplifies to:

$$\lim_{P(p_i=y)\to 1} \mathbb{E}[R_1(r_i)] = 2P(s_i = s_{\text{mode}}|p_i = y) - 1 \tag{24}$$

At this stage, the training objective becomes purely a coordination problem: all correct predictions must agree on the modal score to achieve positive reward. The equilibrium of this coordination game occurs when all correct predictions produce the same score, i.e., when there exists a unique score $s^*$ such that:

$$\pi_\theta(s = s^*|p = y) = 1 \quad \text{and} \quad \pi_\theta(s \neq s^*|p = y) = 0 \tag{25}$$

This implies that $\text{Var}(s_i|p_i = y) = 0$ at equilibrium, achieving perfect consistency among correct predictions.

Define the Lyapunov function $V(\theta) = -\mathbb{E}[R_1(r_i)]$. Along the policy gradient trajectory:

$$\frac{dV}{dt} = -\nabla_\theta \mathbb{E}[R_1(r_i)]^T \frac{d\theta}{dt} \tag{26}$$
$$= -\alpha\|\nabla_\theta \mathbb{E}[R_1(r_i)]\|^2 \leq 0 \tag{27}$$

where $\alpha > 0$ is the learning rate. Since $R_1(r_i)$ is bounded, $V(\theta)$ is bounded below, ensuring convergence to a critical point with the desired properties.

## B.2 Feedback Optimization via Score Improvement

The effectiveness of our Stage 2 training relies on a fundamental insight about the autoregressive generation process and how score improvements can serve as proxies for feedback quality. Our model generates responses following a specific sequential order: error analysis $\to$ evaluation score $\to$ feedback $\to$ final label.

Let $r = (a, s, f, p)$ denote a complete response where $a$ represents error analysis, $s$ the evaluation score, $f$ the feedback, and $p$ the final binary prediction. Under autoregressive factorization, the likelihood decomposes as:

$$\pi_\theta(r|x) = \pi_\theta(a|x)\pi_\theta(s|a, x)\pi_\theta(f|s, a, x)\pi_\theta(p|f, s, a, x) \tag{28}$$

The key observation is that since $s$ is generated before $f$, the evaluation score represents the model's assessment of proof quality based purely on error analysis, independent of the feedback content. This temporal independence allows us to interpret score improvements as objective measures of feedback effectiveness.

Consider the following formalization: let $\tau_0$ be an initial proof attempt and $F_\theta$ our feedback provider. When $F_\theta$ generates feedback $f$, a prover $P$ uses this feedback to produce a revised proof $\tau_1$. If our scoring function is $S_\phi$, then the score improvement is $\Delta s = S_\phi(\tau_1) - S_\phi(\tau_0)$.

The fundamental assumption underlying our approach is that effective feedback should systematically lead to better proofs. Formally, for high-quality feedback $f$, we expect $\mathbb{E}[\Delta s | f] > 0$. This motivates our reward function:

$$R_2(f_i) = \begin{cases} +1 & \text{if } s_{1,i} - s_0 > \delta \\ -1 & \text{if } s_{1,i} - s_0 < -\delta \\ 0 & \text{if } |s_{1,i} - s_0| \leq \delta \end{cases} \tag{29}$$

To understand why this reward structure is optimal, we analyze its expected value. Let $\Delta s_i = s_{1,i} - s_0$ and define the improvement distribution as $P(\Delta s)$. Then:

$$\mathbb{E}[R_2(f_i)] = \int_{\delta}^{\infty} P(\Delta s)\, d(\Delta s) - \int_{-\infty}^{-\delta} P(\Delta s)\, d(\Delta s) \tag{30}$$

$$= P(\Delta s > \delta) - P(\Delta s < -\delta) \tag{31}$$

For this expectation to be positive, we need $P(\Delta s > \delta) > P(\Delta s < -\delta)$, which occurs when the feedback provider generates more improvements than degradations. The policy gradient becomes:

$$\nabla_\theta \mathbb{E}[R_2(f_i)] = \mathbb{E}[R_2(f_i)\nabla_\theta \log \pi_\theta(f_i | s_0, a_0, x)] \tag{32}$$

Substituting our reward structure:

$$\nabla_\theta \mathbb{E}[R_2(f_i)] = \int_{\delta}^{\infty} P(\Delta s)\nabla_\theta \log \pi_\theta(f_i | s_0, a_0, x)\, d(\Delta s) \tag{33}$$

$$- \int_{-\infty}^{-\delta} P(\Delta s)\nabla_\theta \log \pi_\theta(f_i | s_0, a_0, x)\, d(\Delta s) \tag{34}$$

This gradient directly increases the probability of generating feedback that leads to score improvements while decreasing the probability of feedback that causes degradations.

However, a critical challenge emerges from the circular dependency between the feedback provider $F_\theta$ and scorer $S_\phi$. If both components update simultaneously, we encounter a moving target problem. To formalize this instability, consider the joint dynamics of the two components.

At step $t$, the feedback provider parameters $\theta^{(t)}$ are updated based on rewards computed using scorer parameters $\phi^{(t)}$. Simultaneously, if the scorer updates to $\phi^{(t+1)}$, it changes the reward landscape for the next iteration. This creates a coupled dynamical system:

$$\theta^{(t+1)} = \theta^{(t)} + \alpha_\theta \nabla_\theta \mathbb{E}[R_2(f_i; \phi^{(t)})] \tag{35}$$

$$\phi^{(t+1)} = \phi^{(t)} + \alpha_\phi \nabla_\phi \mathbb{E}[R_1(r_j; \theta^{(t)})] \tag{36}$$

The problem is that the reward function $R_2$ depends on $\phi$, so when $\phi$ changes, the reward signal for the same feedback changes, creating instability. To analyze this mathematically, consider the Jacobian of the combined system:

$$J = \begin{bmatrix} \frac{\partial}{\partial \theta}\nabla_\theta \mathbb{E}[R_2] & \frac{\partial}{\partial \phi}\nabla_\theta \mathbb{E}[R_2] \\ \frac{\partial}{\partial \theta}\nabla_\phi \mathbb{E}[R_1] & \frac{\partial}{\partial \phi}\nabla_\phi \mathbb{E}[R_1] \end{bmatrix} \tag{37}$$

The off-diagonal terms $\frac{\partial}{\partial \phi}\nabla_\theta \mathbb{E}[R_2]$ and $\frac{\partial}{\partial \theta}\nabla_\phi \mathbb{E}[R_1]$ represent the coupling between the two optimization problems. When these terms are large, the system can exhibit oscillatory or unstable behavior.

The momentum encoder strategy addresses this by decoupling the update frequencies. Instead of updating both parameters every step, we maintain:

$$\phi^{(t+1)} = \begin{cases} \phi^{(t)} & \text{if } t \bmod m \neq 0 \\ \theta^{(t)} & \text{if } t \bmod m = 0 \end{cases} \tag{38}$$

This creates periods of stability where $\phi$ remains fixed while $\theta$ optimizes against a consistent reward signal. During these intervals, the feedback provider's optimization problem becomes:

$$\max_{\theta} \mathbb{E}[R_2(f_i; \phi_{\text{fixed}})] \tag{39}$$

Since $\phi$ is fixed, the off-diagonal coupling terms vanish, and the optimization becomes stable. The convergence analysis during each fixed-$\phi$ period follows standard policy gradient theory.

When $\phi$ updates (every $m$ steps), it incorporates the improved feedback generation capabilities developed during the previous period. This creates a staircase-like improvement pattern where each plateau represents stable optimization followed by a knowledge transfer step.

To analyze convergence, we need to distinguish between two different measures of performance. Let $Q(\theta)$ represent the true quality of feedback generated by parameters $\theta$, measured by an idealized, consistent evaluation standard. In contrast, $S_{\phi}(\cdot)$ represents the score assigned by the current model parameters $\phi$, which may vary across different parameter settings.

The crucial insight is that while $\theta^{(tm)}$ was optimized to maximize $\mathbb{E}[S_{\phi^{((t-1)m)}}(\tau_{\text{revised}}(\theta))]$, this does not guarantee that $\mathbb{E}[S_{\phi^{(tm)}}(\tau_{\text{revised}}(\theta^{(tm)}))] \geq \mathbb{E}[S_{\phi^{((t-1)m)}}(\tau_{\text{revised}}(\theta^{(tm)}))]$ because the scoring function itself has changed.

However, we can establish convergence through a different approach. Consider the sequence of feedback quality improvements measured by a fixed, external evaluation standard $Q^*(\cdot)$. During each interval $[(t-1)m, tm)$, the feedback provider $\theta$ is optimized according to:

$$\theta^{(k+1)} = \theta^{(k)} + \alpha \nabla_{\theta} \mathbb{E}[R_2(f; \phi^{((t-1)m)})] \tag{40}$$

Since the reward $R_2$ is designed to correlate with true improvement (i.e., $\mathbb{E}[R_2(f; \phi)] > 0$ when $Q^*(f) > Q^*(\text{baseline})$), we have:

$$Q^*(\theta^{(tm)}) \geq Q^*(\theta^{((t-1)m)}) \tag{41}$$

This inequality holds because $\theta^{(tm)}$ was specifically trained to generate feedback that leads to improvements as measured by a scorer that was previously optimized for the same objective.

The momentum update ensures that the new scorer $S_{\theta^{(tm)}}$ inherits the improved capabilities from the feedback training process. While we cannot guarantee that the numerical scores will increase, we can establish that the overall system capability improves monotonically.

To formalize this, define the system-wide performance as:

$$\Phi(t) = \max_{\tau} \mathbb{E}[Q^*(\tau)|\tau = P(F_{\theta^{(t)}}(\tau_0), \tau_0)] \tag{42}$$

This represents the best possible proof that can be achieved by applying feedback from the current model. Under our training scheme:

$$\Phi(tm) \geq \Phi((t-1)m) \tag{43}$$

The momentum update preserves this monotonic improvement while providing training stability. The key insight is that even though individual score values may fluctuate due to changing evaluation criteria, the underlying capability to generate effective feedback improves consistently.

## C  TRAINING APPROACH DISCUSSION

The backbone of PROOF-VERIFIER is Qwen3-8B. Our model is trained exclusively using RL without prior SFT, as we find that additional SFT training reduces the diversity of the exploration space during rollout generation while providing only marginal performance improvements. We identify two main advantages for score distribution of SFT through pre-experiments: (1) distilling knowledge from larger models (e.g., Qwen3-235B) to smaller models improves the robustness of evaluation score distributions, and (2) applying self-consistency filtering strategies to datasets generated by the model itself for self-training also enhances robustness. However, we find that these benefits

can be naturally integrated into the RL training process by designing reward function that provids positive rewards only for responses that are both correct and group-consistent. Additionally, while SFT traditionally is used to establish output format, this can be achieved directly through the reward function design. Thus, PROOF-VERIFIER is trained using GRPO entirely, with the reward function serving as the core design mechanism. The training process consists of two steps, detailed below, following a coarse-grained to fine-grained training objective progression.

## C.1 EVALUATION STRATEGY DESIGN

When performing multiple evaluations on a single proof candidate, the stability and reliability of evaluation scores are crucial for robust assessment. We analyze two key factors that influence evaluation quality: (1) sampling strategy, including Single Sample, Parallel Sampling, and IID sampling, where the key distinction between the latter two is that parallel sampling generates trajectories that influence each other through various parameters (e.g., repetition penalties, beam group sizes), while IID sampling generates completely independent evaluations, and (2) prompt complexity, comparing simple prompts versus detailed rubric-based prompts. We conduct experiments across multiple benchmarks including CombiBench, FIMO, miniF2F, ProofNet, Prover-Bench, and PutnamBench. Using LLM-based provers, we first generate informal and formal proof candidates via IID sampling, then evaluate them using two models of different scales: Qwen3-8B and Qwen3-235B. Results are shown in Table 3, where outliers represent abnormal scores exceeding the expected [0-100] range, ICC measures the consistency of ratings across multiple evaluations of the same proof, and Median CV quantifies the relative variability in evaluation scores.

We find that parallel sampling produces more stable evaluations than IID sampling, with lower outlier rates and higher consistency metrics. Additionally, parallel sampling achieves higher computational efficiency under VLLM-optimized infrastructure. Regarding prompt design, complex prompts containing detailed scoring rubrics surprisingly underperform compared to simple, direct prompts, likely due to increased instruction complexity leading to inconsistent interpretation. Therefore, we adopt Simple Prompt + Parallel Sampling for all subsequent experiments.

| Sampling | Median CV ↓ | ICC ↑ | Outlier(%) ↓ |
|---|---|---|---|
| **Qwen3-8B** | | | |
| C+Parallel | 0.1031 | 0.5540 | 4.3 |
| ↪ IID | 0.1056 | 0.5537 | 4.7 |
| S+Parallel | _0.0979_ | _0.6140_ | _1.5_ |
| ↪ IID | 0.0983 | 0.5737 | 1.6 |
| **Qwen3-235B** | | | |
| C+Parallel | 0.0770 | 0.8085 | 1.4 |
| ↪ IID | 0.0769 | 0.8081 | 1.6 |
| S+Parallel | **0.0713** | **0.8372** | **0.2** |
| ↪ IID | 0.0753 | 0.8128 | 0.3 |

Table 3: Consistency and reliability of evaluation scores across different models, prompts, and sampling strategies. S/C denotes simple/complex prompts, respectively.

# D HUMAN ANNOTATION INTERFACE

We developed a web-based interface for human evaluation of mathematical proof assistance quality through pairwise comparisons within each response type.

## D.1 INTERFACE DESIGN

The interface displays four components (Figure 7): the original mathematical statement at the top, the student's proof attempt below it, two response panels side-by-side in the center, and selection buttons at the bottom. Each comparison presents two responses of the same type (e.g., error analysis A vs. error analysis B).

## D.2 EVALUATION SETUP

For each response type, annotators complete 20 pairwise comparisons. Panel positions are randomized to avoid bias. Annotators select which response would be more helpful for a student.

**Mathematical Statement**

Given positive integers $n$ and $k$, there is a finite set $A(n,k) = \{\alpha_1, \alpha_2, ..., \alpha_n\}$ where $\alpha_i = (a_{i,1}, a_{i,2}, ..., a_{i,(j-1)k+1})$, $a_{i,1}, a_{i,2}, ..., a_{i,(j-1)k+1} \in \{0, 1\}$ and $a_{i,1} + a_{i,2} + ... + a_{i,(j-1)k+1} = 1$ for $i = 1, 2, ..., n$. For an $n$-term sequence $\Omega : (1, x_1), (2, x_2), ..., (n, x_n)$, if $x_1 = 1$ and $x_{i+1} - x_i \in \{0, 1, ..., k\}$ for $i = 1, 2, ..., n - 1$, then it is called a "G-sequence". At this time, denote $G(\Omega) = a_{1,x1} + a_{2,x2} + ... + a_{n,xn}$. Prove that for any set $A(n,k)$, there exists a "G-sequence" $\Omega$ such that $(k + 1)G(\Omega) > kn$.

**Student's Proof Attempt**

**Proof:** I will use the probabilistic method to show the existence of such a G-sequence.

First, let me count the total number of G-sequences. Since $x_1 = 1$ and each subsequent $x_{i+1}$ can increase by at most $k$, we have at most $(k+1)^{n-1}$ possible G-sequences.

Now I'll compute the sum $\sum G(\Omega)$ over all G-sequences $\Omega$. For each position $i$ and value $j$, the term $a_{i,j}$ appears in exactly the number of G-sequences where $x_i = j$. Since $\sum_j a_{i,j} = 1$ for each $i$, and each G-sequence visits exactly one position per step, we get $\sum G(\Omega) \geq$ (number of G-sequences).

By the pigeonhole principle, there exists a G-sequence $\Omega$ such that $G(\Omega) \geq$ (average value). Since the total sum is at least the number of sequences, the average is at least 1.

Therefore, there exists a G-sequence with $G(\Omega) \geq 1$, which gives us $(k+1)G(\Omega) \geq k+1 > kn$ when $k+1 > kn$. *However, this argument breaks down when $k+1 \leq kn$...*

**Feedback A - Error Analysis**

The proof attempt demonstrates an understanding of the probabilistic method and the pigeonhole principle but contains critical errors in the application of these concepts. Here's a breakdown of the feedback:

**1. Conceptual Understanding**: The user correctly identifies the total number of G-sequences and the structure of $G(\Omega)$. They also attempt to use the probabilistic method and the pigeonhole principle, which are valid strategies for such problems.

**2. Logical Gaps**: The core error lies in the incorrect inequality $(k + 1)^n > kn \cdot (k + 1)^{n-1}$. This inequality is not generally true, and the proof fails to address cases where it does not hold (e.g., $n = 2$, $k = 1$). The argument also misapplies the pigeonhole principle by assuming that the average value being greater than $kn$ guarantees the existence of a sequence exceeding $kn$.

**3. Suggestions for Improvement**:
- **Re-examine the Inequality**: The proof should address the general case by considering the relationship between $k + 1$ and $kn$ for all $n$ and $k$. For example, when $k + 1 \leq kn$, alternative methods (e.g., explicit construction of a G-sequence) may be necessary.
- **Clarify the Pigeonhole Principle Application**: The argument should explicitly show that the average value of $(k + 1)G(\Omega)$ being at least $k + 1$ ensures the existence of a sequence with $(k + 1)G(\Omega) > kn$, which requires a more rigorous analysis of the total sum and individual contributions.
- **Consider Edge Cases**: The proof should test edge cases like $n = 1$ or $k = 1$ to ensure the result holds universally.

**Feedback B - Actionable Feedback**

The proof shows promise in its approach but requires significant structural improvements to establish the desired inequality. Here's a constructive analysis:

**Strengths Identified**: You correctly recognize that this is an extremal problem requiring either probabilistic or constructive methods. The notation is properly established, and you understand the constraint structure of G-sequences.

**Critical Issues to Address**:
- **Mathematical Rigor**: The current argument lacks the necessary inequality chain. You need to establish that max $G(\Omega) \geq$ average $G(\Omega)$, then show this average exceeds $kn/(k+1)$.
- **Counting Analysis**: Properly enumerate all possible G-sequences and their contributions. The total number is at most $(k+1)^{n-1}$, not $(k+1)^n$.
- **Boundary Cases**: Verify your approach works for small values: $n=1,k=1$ gives trivial satisfaction, but $n=2,k=1$ requires careful analysis.

**Recommended Next Steps**:
1. **Establish Lower Bound**: Prove that $\sum G(\Omega) \geq n \cdot$ (total sequences), using the constraint that each position contributes at least once across all sequences.
2. **Apply Extremal Principle**: Since the maximum value must be at least the average, show max $G(\Omega) \geq n \cdot (k+1)^{n-1} > kn/(k+1)$.
3. **Verify Inequality**: Confirm that $n/(k+1) > kn/(k+1)$ simplifies to the target inequality.

This approach transforms your intuitive understanding into a rigorous proof framework.

**Which feedback is more helpful?**

[ Choose A ]    [ Choose B ]

You selected Feedback B as more helpful!
Selection recorded at 11:33:11 PM

Figure 7: Human annotation interface for pairwise comparison of mathematical proof assistance types. The web-based evaluation tool presents annotators with a mathematical statement and two alternative response types from three categories: error analysis, actionable feedback, and refined proof.

### D.3 Win Rate Analysis

After collecting preferences across all pairs, we analyze correlations between win rates within the same verification trajectory. For example, if verification trajectory A's error analysis outperforms trajectory B's error analysis, we examine whether trajectory A's actionable feedback also outperforms trajectory B's feedback, revealing consistency across response types.

## E  Related Work

**Mathematical theorem proving** has seen significant progress with benchmarks like miniF2F (Zheng et al., 2022), FIMO (Liu et al., 2023), and PutnamBench (Tsoukalas et al., 2024), alongside advances in AI-assisted approaches such as AlphaGeometry (Trinh et al., 2024; Chervonyi et al., 2025) and recent neural theorem provers (Polu & Sutskever, 2020). Current verification relies primarily on interactive theorem provers like Lean4 (de Moura et al., 2015; de Moura & Ullrich, 2021), Coq (Huet & Paulin-Mohring, 2000), and Isabelle (Nipkow et al., 2002), with recent improvements in compilation efficiency through Kimina Lean Server (Santos et al., 2025). However, these approaches have critical limitations for RLVR applications: they only support formal languages while excluding natural language proofs, and provide only binary verification results without the detailed error analysis or actionable feedback essential for effective reinforcement learning.

**Reward modeling for mathematical reasoning** has primarily focused on traditional problem-solving tasks. Process reward models (Lightman et al., 2024; Wang et al., 2024a) have shown success in step-by-step verification for computational problems, while outcome reward models achieve strong results through final answer comparison (Cobbe et al., 2021). More broadly, LLM-as-a-judge approaches (Zheng et al., 2023) have evolved from instruction-following evaluation to reward models across domains including medical QA (Croxford et al., 2025), multimodal tasks (Chen et al., 2024), and code generation (Zhao et al., 2024), with open-source alternatives like PandaLM (Wang et al., 2024b) and Prometheus (Kim et al., 2024a;b) providing cost-effective solutions. While these methods succeed in traditional mathematical tasks through reference answer comparison (Chen et al., 2025), theorem proving's process-oriented nature and diverse solution paths present unique verification challenges that limit RLVR applications in this domain. Our work addresses this verification bottleneck by developing the first dual-language verifier specifically designed for mathematical theorem proving, enabling RLVR applications through detailed error analysis and fine-grained evaluation capabilities via novel two-stage reward modeling.

## F  Training Dataset Statistics

Our initial data is divided into natural language data and formal language data, including the statement-proof pairs labeled to be correct or not.

For statement-proof pairs in natural language, we utilize the statements from OPC dateset Dekoninck et al. (2025) and RFM dataset Guo et al. (2025). We use different reasoning models to generate multiple proofs for a single statement, which we then label as either correct or incorrect. Of the statement-proof pairs in our initial pool, 2,000 are from the RFM dataset (582 labeled as correct, 1,418 as incorrect) and 3,039 are from the OPC dataset (1,109 labeled as correct, 1,930 as incorrect). To balance the dataset, we remove a number of incorrect proofs. The final dataset consists of 1,164 entries from the RFM dataset and 2,218 from the OPC dataset, with an equal number of incorrect and correct proofs. We evenly extract data from the formal statement-proof pairs of datasets including MiniF2F Zheng et al. (2022), ProofNet Azerbayev et al. (2023), PutnamBench Tsoukalas et al. (2024), ProverBench Ren et al. (2025), CombiBench Liu et al. (2025), Fimo Liu et al. (2023), and Hmmt Zhang et al. (2025b) (Note that the number of statements in these datasets varies, and the quantity of our dataset refers to the number of statement-proof pairs but not statements. Besides, dataset like Hmmt contains statements and proofs in natural language, and we translate them into formal language), and ensure that the total amount of formal data is roughly the same as that of natural language data. Ultimately, we obtain a final dataset of 6764 entries, as shown in Table 4. And we also count the distribution of knowledge domains, as shown in Table 5, which can be seen as a rough evaluation of the diversity of our dataset.

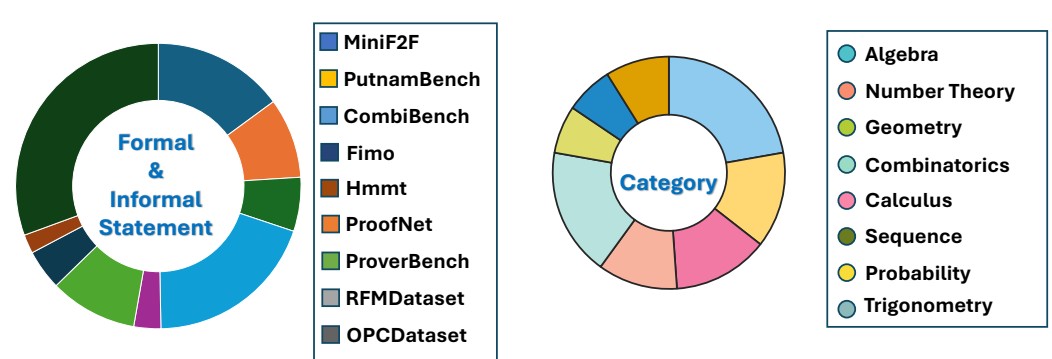

Figure 8: Benchmark statistics and category distribution.

| Dataset Name | Data Size |
|---|---|
| OPC Dataset | 2218 |
| RFM Dataset | 1164 |
| MiniF2F | 484 |
| ProofNet | 484 |
| PutnamBench | 484 |
| ProverBench | 484 |
| CombiBench | 484 |
| Fimo | 484 |
| Hmmt | 484 |
| Total | 6770 |

Table 4: Statistics of training set data sources.

## G  PROMPT TEMPLATE DESIGN

These training datasets are formatted into a predefined prompt template for training. The evaluator is instructed to generate error analysis, evaluation scores, and actionable feedback as output. Natural language inputs include the statement and proof, while formal language inputs additionally include compilation results. We find this essential because without compilation results, the trained evaluator often produces incorrect analyses with low accuracy. This occurs because Lean4 proofs can encompass numerous proof steps and strategies within just one or two lines of code, requiring significant compilation time even for the compiler. Rather than having the evaluator perform lengthy reasoning to interpret the code and analyze potential errors, we directly provide compilation results to help it better understand the code and proof states, enabling more accurate and targeted feedback.

We compared two types of prompts: 1) simple free-style prompts that only constrain the format of model outputs, versus 2) complex prompts with specific guidance, such as requiring analysis from given perspectives, summing scores across multiple dimensions, and generating feedback from pre-defined frameworks. These two types of prompts are shown in Appendix M. Counterintuitively, we found that the first prompt yields significantly better results than the second. Even large models may make errors when summing evaluation scores and struggle to follow complex instructions. For the second type, models fail to follow the instruction to provide a detailed and in-depth analysis for each required perspective, instead offering only superficial and cursory responses. In contrast, simple prompts allow models to perform targeted, self-adaptive detailed analysis and provide specific feedback. For training, overly specific and complex prompts can be viewed as parameterized prefixes that constrain the exploration space of conditional generation rollouts, thereby preventing model improvement. Simple prompts provide models with sufficient search space, where as long as the format is correct, the optimization of intermediate processes relies on self-exploration, which

| Domain | Data Size |
|--------|-----------|
| Algebra | 1480 |
| Number Theory | 1184 |
| Calculus | 958 |
| Geometry | 886 |
| Combinatorics | 740 |
| Sequence | 592 |
| Probability | 392 |
| Trigonometry | 422 |

Table 5: Statistics of different mathematical domains.

can fully leverage the strengths of RL to train more powerful models. Our experiments confirm this point: in RL, specific and complex prompts not only perform poorly initially but also provide limited improvement for the model. Detailed experimental results are shown in Appendix K.

## H   RFM DATASET

The prompt template follows a rubric-based evaluation containing 10 perspectives, covering various common proof errors such as Transformation Error, Hidden Assumption, and Boundary Neglect, as detailed in Appendix M. We employed three PhD-level math students to independently label the data (discarding entries where consensus was not reached), and comparison revealed a Pearson correlation coefficient of 0.89, thus demonstrating the feasibility of using this approach for large-scale labeling of this dataset, where the noise level remains within acceptable bounds.

**1) OPC Dataset** Dekoninck et al. (2025): Contains proof attempts generated by multiple advanced reasoning models using natural language on PutnamBench (Tsoukalas et al., 2024) statements. Multiple human experts label these proof attempts as True or False. For cases where annotators disagree, we apply majority voting when feasible; otherwise, we discard instances with substantial disagreement.

**2) RFM Bench** Guo et al. (2025): A benchmark contains proof statements across multiple difficulty levels (high school, undergraduate, graduate) and mathematical domains, including geometry, algebra, inequalities, number theory, and calculus. All statements are human-annotated and verified for quality and difficulty control. Since this dataset contains only statements, we generate proof candidates using advanced reasoning models and employ LLM-as-a-judge labeling with Gemini-2.5-Pro as the evaluator, as we find that Gemini-2.5-Pro demonstrates strong proof evaluation capabilities on this dataset under carefully designed rubrics. Finally, we extract 100 {statement, proof attempt} pairs for human annotation to form part of our test set.

Both datasets may introduce validation errors, potentially overlooking logical issues and other subtle errors that are difficult to detect. Moreover, they contain only natural language proofs without formal language coverage. To address these limitations, we introduce a third data construction method.

## I   TEST DATASET COLLECTION

For formal language, we selected different prover models (DeepSeek Prover V2 Ren et al. (2025), Kimina Prover Wang et al. (2025a), Goedel Prover Lin et al. (2025), Leanabell Prover Zhang et al. (2025a), and STP Prover Dong & Ma (2025)) that differ from training data collection models, sampling 32 proof attempts per statement on new benchmarks with compiler-generated labels. For natural language, we used various series models (Qwen3, Mistral, Magistral Mistral-AI et al. (2025), Qwen 2.5, Gemma Team et al. (2025), DeepSeek R1) for sampling, with 100 samples labeled by human annotators and the remaining labeled by Gemini-2.5-Pro. Our three human annotators achieved Cohen's Kappa Cohen (1960) scores of 0.86, 0.86, and 0.88, validating the high quality and reliability of the test set. The final test set contains 5k formal language and 1k natural language samples.

## J MODEL INTRODUCTION

### J.1 GENERAL PROPOSE MODELS

**Qwen series.** We use the Qwen-2.5-7B, Qwen-2.5-72B, Qwen-3-8B and Qwen3-30B Qwen et al. (2025) in our experiments. Both Qwen2.5-8B and Qwen3-8B acquire extensive mathematical knowledge during pre-training, and Qwen3-8B, in particular, has ability to handle more difficult mathematical problems. As a result, they are widely used as base models for training in research. Qwen-2.5-72B is the largest open-source model in Qwen-2.5 series, which demonstrates significant improvements in mathematical ability compared to its predecessors. Qwen3-30B, a new representative model of the Qwen series, adopts a mixture-of-experts architecture and can employ long chain-of-thoughts, which greatly improves its mathematical reasoning capability. Specifically, the Qwen3-30B model shows strong performance on competition-level math benchmarks, such as AIME-2024 Jia (2025) and AIME-2025 OpenCompass Team (2025).

**Mistral and Magistral series.** Mistral Mistral-AI et al. (2025) is a series of powerful, efficient language models from the Mistral AI. The models are known for their strong performance across various tasks while being efficient. These models achieve good performance on various mathematical benchmarks like MATH Hendrycks et al. (2021) and GSM8K Cobbe et al. (2021), proving their strong capabilities from basic arithmetic to advanced problem-solving. To further improve models' reasoning ability, Mistral AI introduced the Magistral series. These new models are specifically trained for advanced reasoning tasks and capable of performing long chain-of-thought. This makes their problem-solving process more transparent and reliable for complex applications, while achieving high scores on challenging math benchmarks like AIME-2024 Jia (2025) and AIME-2025 OpenCompass Team (2025).

**Gemma series.** Gemma Team et al. (2025) is a family of lightweight, open-source models developed by Google. The recent Gemma-3 series represents a significant leap forward in capabilities and efficiency. A key highlight of the Gemma 3 series is its exceptional performance in mathematical reasoning. The technical report confirms that Gemma 3 models demonstrate substantial gains over their predecessors on critical math benchmarks, specifically MATH and HiddenMath.

**Deepseek-R1.** Deepseek-R1 DeepSeek-AI et al. (2025) is one of the most prominent open-source reasoning models, widely used for complex mathematical reasoning. Its performance on two competition-level datasets, AIME-2024 and AIME-2025, is comparable to top commercial models, achieving a pass@1 score of nearly 90% or higher. Furthermore, Deepseek-R1's recent performance on the RFM Dataset for mathematical proofs is shown to surpass many reasoning models.

**Gemini-2.5-pro.** The Gemini-2.5-pro Comanici et al. (2025) model is dedicated to pushing the frontier of AI with advanced reasoning. In addition to its excellent performance on common math competition datasets like AIME, this model has also recently been shown to surpass other models in its mathematical proof generation ability under the same criteria.

### J.2 PROVER MODELS

In this paper, we refer to models that are specifically trained to generate formal proofs from formal statements as prover models. In this subsection, we provide a short introduction to the prover models we used.

**DeepSeek-Prover-V2.** DeepSeek-Prover-V2 Ren et al. (2025) is an open source prover model for theorem proving in Lean 4, which is distinguished by its innovative pipeline that effectively unifies informal, human-like reasoning with the rigor of formal proof generation. Its core feature is a two-stage process that begins by using a powerful general model (DeepSeek-V3 DeepSeek-AI et al. (2025)) to decompose complex theorems into a high-level proof sketch composed of simpler subgoals. These subgoals are then solved by a more specialized prover model.

**KIMINA Prover.** KIMINA Prover Wang et al. (2025a) emulates human problem solving by generating a mix of informal mathematical intuition and formal Lean 4 code. This process allows it to iteratively build and refine a proof.

**Goedel Prover.** The core innovation of Goedel Prover Lin et al. (2025) lies in addressing the scarcity of formal mathematical data through a two-pronged approach. It automatically translates a massive

dataset of 1.64 million math problems from natural language into formal statements. Besides, it employs a training method where the model generates proofs for these statements, and any new correct proofs are added back into the training set to iteratively create a more powerful prover.

**Leanabell Prover.** Leanabell Prover Zhang et al. (2025a)undergoes a two-stage post-training strategy. The first stage is supervised fine-tuning on a large, custom-curated dataset. This dataset not only includes a massive collection of statement-proof pairs but, more importantly, incorporates synthetic data designed to integrate self-reflection and error correction. This is achieved by generating Chain-of-Thought style explanations for why a proof failed and how it is corrected. Second, using the GRPO algorithm, the finetuned model learns by generating entire proofs and receiving a direct reward signal from the Lean 4 compiler based on whether the proof is successfully verified. This strategy of combining SFT with cognitive data and RL optimization allows Leanabell-Prover to achieve good performance.

**STP Prover.** The training of STP prover Dong & Ma (2025) simultaneously operates in two roles of prover and conjecturer. These two roles create a dynamic self-play feedback loop. The conjecturer creates novel problems that are barely provable by the prover's current ability. This process generates a continuous stream of appropriately difficult training data. This allows the model to improve its reasoning skills without requiring more human-created datasets, effectively creating its own adaptive learning curriculum.

## K    ABLATION STUDY ON PROMPT DESIGN

| Verifier | Acc | Prec | Rec | F1 |
|---|---|---|---|---|
| Simple Prompt | 0.57 | 0.58 | 0.55 | 0.56 |
| Simple Prompt + RL | 0.86 | 0.87 | 0.85 | 0.86 |
| Complex Prompt | 0.43 | 0.45 | 0.42 | 0.43 |
| Complex Prompt + RL | 0.52 | 0.53 | 0.51 | 0.52 |

Table 6: Comparison of prompt types during inference and after RL training. Simple Prompt outperforms Complex Prompt in both settings.

As shown in Table 6, we found that Simple Prompt performs better than Complex Prompt in both zero-shot stage and after RL training. The reason is that overly complex prompts are difficult for models to follow, resulting in poor inference performance. Additionally, overly complex prompts limit the model's exploration space, where model outputs become constrained and RL cannot effectively optimize the model further.

## L    TRANSLATION ERROR

In this section, we provide cases to show the errors that occurred when translating the formal statements or proofs to natural language statements or proofs.

### L.1    STATEMENT TRANSFORMATION ERROR

We identify three main kinds of statement transformation errors: *Premise Difference*, *Conclusion Difference*, *Not to Prove*. *Premise Difference* refers to a discrepancy where the premise provided by the translated natural language statement differs from those off the original formal statement,as shown in Figure 9, 10, 11. In each figure, we mark the translation errors in red and provided an analysis of the errors.Taking Figure 9 as an example, the original formal statement does not provide aby specific value for the function f(x) and its corresponding inverse function at a given point, but only presented an equality condition. However, the translated natural language statement does include a value. Although these values do not affect the solution in this specific example, it still highlights a semantic inconsistency.

*Conclusion Difference* refers to a discrepancy where the conclusion to be proven in the formal language statement differs from the conclusion of the translated natural language statement, as shown

in Figure 10, 11, 12, 13, 14. For example, in Figure 12, the conclusion to be proven in the formal language statement is that a certain property holds for all positive integers. However, the natural language statement is incorrectly rendered as proving there are infinitely many such integers. While these two concepts are somewhat related, the difference between them is significant.

*Not to Prove* refers to a situation where the natural language statement fails to clearly distinguish between the premises and the conclusion to be proven(Figure 15, 16). In some cases, the translation is even a problem that requires a numerical result instead of a proof(Figure 17).

---

**Formal Statement and proof:**

```
import Mathlib
import Aesop
set_option maxHeartbeats 0
open BigOperators Real Nat Topology Rat
/-- Suppose f(x) is an invertible function with f(2)=f^{-1}(2)=4.
    Show f(f(2)) = 2. -/
theorem mathd_algebra_188 (σ :  Equiv R R)
    (h :  σ^{-1} 2 = σ 2) :
    σ (σ 2) = 2 := by
```

**Transformed Natural Language Statement:**

Suppose $f$ is a bijective function from the real numbers to the real numbers, with the property that both $f(2)$ and its inverse function $f^{-1}(2)$ evaluate to 4. What is the value of $f(f(2))$? Show that it equals 2.

**Error Type:** Premise Difference
**Error Analysis:** The natural language statement describes the specific values $f(2) = 4$ and $f^{-1}(2) = 4$, while the Lean code states that $f(2)$ and $f^{-1}(2)$ are equal, without explicitly mentioning the value 4. This error may have been affected by the comment content in the lean code.

---

Figure 9: A failure case of *Premise Difference*.

## L.2 PROOF TRANSLATION ERROR

We also explored the bad cases of converting formal proofs into natural language proofs. A common case is that the natural language proof translated by the model is merely an annotation of the formal proof, as shown in Figure 18 and 19. This means it is not a complete proof on its own, because languages like Lean 4 contain elements such as tactics that are not fully explained in the translation. Furthermore, we also discover cases where the natural language proofs and formal proofs use different methods(Figure 20). Although these proofs may be correct on their own, they fail to meet our consistency requirement.

## M   PROMPTS

In this section, we detail the prompts used in our experiments.

Figure 21 is the prompt we provide to *Gemini-2.5-pro* to judge the proofs generated by various LLMs for statements from RFM Dataset. This prompt not only asks the model to provide an overall correctness score, but also requires it to categorize any errors into one or more failure modes, which are detailed in Figure 22.

Figure 23 and 24 show the prompts provided to our evaluator to evaluate a formalized proof, with and without verification information from the Lean 4 compiler, respectively.

Figure 25 and 27 show the prompts provided to our evaluator to evaluate a natural language proof, with and without detailed rubrics to give the final score, respectively.

**Formal Statement and proof:**

```
import Mathlib
open Matrix
def coeff_matrix : Matrix (Fin 4) (Fin 3) Q :=
  !![!![ -19,   8,   0],
      ![ -71,  30,   0],
      ![  -2,   1,   0],
      ![   4,   0,   1]]
def aug_matrix : Matrix (Fin 4) (Fin 4) Q :=
  !![!![ -19,   8,   0, -108],
      ![ -71,  30,   0, -404],
      ![  -2,   1,   0,  -12],
      ![   4,   0,   1,   14]]
axiom rank_of_matrix :   rank coeff_matrix = 2 ∧ rank aug_matrix = 2
axiom verify_solution :   (4, -4, -2) ∈ {x :   Q × Q × Q |
-19 * x.1 + 8 * x.2.1 = -108 ∧
-71 * x.1 + 30 * x.2.1 = -404 ∧
-2 * x.1 + x.2.1 = -12 ∧
4 * x.1 + x.2.2 = 14}
/-- The system of linear equations has a unique solution (4, -4, -2).
    -/
theorem unique_solution : ∃! x : Q × Q × Q,
  -19 * x.1 + 8 * x.2.1 = -108 ∧
  -71 * x.1 + 30 * x.2.1 = -404 ∧
  -2 * x.1 + x.2.1 = -12 ∧
  4 * x.1 + x.2.2 = 14 := by
```

**Transformed Natural Language Statement:**

Consider the system of linear equations:
1) -19x + 8y = -108
2) -71x + 30y = -404
3) -2x + y = -12
4) 4x + z = 14
Prove that this system has a unique solution (4, -4, -2).

**Error Type:** Premise Difference and Conclusion Difference
**Error Analysis:** In the formalized statement, it is directly given that the rank of the coefficient matrix is equal to the rank of the augmented matrix, which falls under the category of premise difference. The formalized statement directly indicates that (4, -4, 2) is a solution to the equation, so it only needs to prove uniqueness. However, the natural language statement first needs to prove that (4, -4, 2) is a solution to the equation, which belongs to conclusion difference.

Figure 10: A failure case of *Premise Difference* and *Conclusion Difference*.

## N  FEEDBACK CASE STUDY

To intuitively demonstrate the effectiveness of our evaluator, we present some cases in this section, as shown in Figures 28, 29, 30, and 31. Figures 28, 29, and 30 show feedback on formal proofs, while Figure 31 provides feedback on a natural language proof.

From the feedback on formal proofs, we can clearly see that the evaluator's feedback can identify and correct errors related to the properties of Lean, as well as point out the missing parts in the formal proof. For example, the case shown in Figure 30 indicated that the feedback precisely identifies the critical gap, which is the missing proof that $\pi < 4$ required to apply the function's conditional definition—and resolves it by directing the user to the essential Mathlib theorem Real.pi_lt_four. It corrects the ineffective simp_all tactic by prescribing explicit hypothesis applica-

**Formal Statement and proof:**

```
import Mathlib
/-- Let f(u,v) be differentiable with
    ∂f/∂u − ∂f/∂v = 2(u−v)e^{−(u+v)}
    and f(u,0) = u²e^{−u}. Then:
    a) For g(x,y) = f(x,y−x) we have ∂g/∂x = (4x−2y)e^{−y};
    b) f(u,v) = (u²+v²)e^{−(u+v)};
    c) f has a unique global minimum at (0,0) with f(0,0) = 0. -/
open Real
variable {u v x y :  R }
noncomputable def f (u v :  R) : R) := (u² + v²) * exp (-(u + v))
noncomputable def g (x y :  R) : R) := f x (y − x)
axiom f_partial_condition (u v :  R) :
deriv (fun u => f u v) u − deriv (fun v => f u v) v = 2 * (u − v) *
    exp (−(u+v))
axiom f_initial_condition (u :  R)) :
f u 0 = u² * exp (−u)
axiom partial_g_x (x y :  R) :
deriv (fun x => g x y) x = (4 * x − 2 * y) * exp (−y)
lemma g_expression (x y :  R) :
g x y = (x² + (y − x)²) * exp (−y) :=
```

**Transformed Natural Language Statement:**

Consider a differentiable function $f(u,v)$ that satisfies:
1. The difference between its partial derivative with respect to $u$ and its partial derivative with respect to $v$ equals $2(u-v)e^{-(u+v)}$
2. The initial condition $f(u,0) = u^2 e^{-u}$
We are to show that:
a) For $g(x,y) = f(x,y-x)$, the partial derivative of $g$ with respect to $x$ is $(4x-2y)e^{-y}$
b) The function $f(u,v)$ has the explicit form $(u^2+v^2)e^{-(u+v)}$
c) The function $f$ has a unique global minimum at $(0,0)$ with value 0

**Error Type:** Premise Difference and Conclusion Difference
**Error Analysis:** In the formalized statement, it is only necessary to prove that g(x, y) = (x² + (y - x)²) * exp(-y). However, the natural language statement needs to prove three statements. Among them, statements (a) and (b) are given as known conditions in the formalized statement, and statement (c) is not reflected in the lean code. Therefore, it falls into both the categories of *Premise Difference* and *Conclusion Difference*.

Figure 11: A failure case of *Premise Difference and Conclusion Difference*

tion ($h_1$ _ ⟨Real.pi_nonneg, Real.pi_lt_four⟩), transforming an incomplete automation attempt into a robust, verifiable proof that adheres strictly to the function's domain conditions.

From the feedback on natural language proofs shown in Figure 31, we can see the feedback accurately point out that the inequality used in the proof is invalid and does not hold under certain conditions. It even provided failure cases of the inequality and a corresponding improvement plan.

**Formal Statement and proof:**

```
import Mathlib
import Aesop
set_option maxHeartbeats 0
open BigOperators Real Nat Topology Rat
/-- For how many positive integers m does there exist at least one
    positive integer n
such that m · n ≤ m + n?
(A) 4    (B) 6    (C) 9    (D) 12    (E) infinitely
    many. Show that it is (E) infinitely many.-/
theorem amc12a_2002_p6 (n : N)
    (h_0 :   0 < n) :
    ∃m, m > n ∧ ∃p, m * p ≤ m + p  := by
```

**Transformed Natural Language Statement:**

How many positive integers $m$ exist such that there is at least one positive integer $n$ satisfying the inequality $m \cdot n \leq m + n$? The options are: - (A) 4 - (B) 6 - (C) 9 - (D) 12 - (E) infinitely many. We are to show that the correct answer is (E) infinitely many.

**Error Type:** Conclusion Difference
**Error Analysis:** The formalized statement means that a certain conclusion holds for any $n$ greater than 0, while the natural language statement means that there are infinitely many values of $n$ for which this conclusion holds. Therefore, this belongs to a conclusion difference.

Figure 12: A failure case of *Conclusion Difference*

**Formal Statement and proof:**

```
import Mathlib
/-- Let A be a real 2 × 2 matrix. Then the determinant of A² is always
    nonnegative, i.e., det(A²) ≥ 0.-/
open Matrix
variable (A : Matrix (Fin 2) (Fin 2) R)
lemma det_A_sq : det (A * A) = (det A) ∧ 2 := by
```

**Transformed Natural Language Statement:**

Let $A$ be a real $2 \times 2$ matrix. Show that the determinant of $A^2$ is always non-negative, i.e., $\det(A^2) \geq 0$.

**Error Type:** Conclusion Difference
**Error Analysis:** In the formalized statement, it is required to prove that the determinant of a matrix squared is equal to the square of the determinant. However, the natural language statement needs to prove that the determinant of a matrix squared is greater than or equal to 0. Therefore, this belongs to a Conclusion Difference.

Figure 13: A failure case of *Conclusion Difference*.

**Formal Statement and proof:**

```
import Mathlib
import Aesop
set_option maxHeartbeats 0
open BigOperators Real Nat Topology Rat

/-- What is the tens digit of 5^2005? Show that it is 2. -/
theorem mathd_numbertheory_198 :
  5 ^ 2005 % 100 = 25 := by
```

**Transformed Natural Language Statement:**

What is the tens digit of $5^{2005}$? Prove that it is 2.

**Error Type:** Conclusion Difference
**Error Analysis:** The statement in the formal language requires proving that the remainder is 25, while the statement in natural language only needs to prove that the tens digit of the remainder is 2.

Figure 14: A failure case of *Conclusion Difference*.

**Formal Statement and proof:**

```
import Mathlib
/- The composition of the functions u(x) = sin x and v(x) = x² is
    u(v(x)) = sin(x²). -/
open Real
noncomputable def u (x : R) : R := sin x
def v (x : R) : R := x²
theorem composition_of_sine_and_quadratic (x : R) :
  u (v x) = sin(x²) := by
```

**Transformed Natural Language Statement:**

What is the composition of the functions $u(x) = \sin x$ and $v(x) = x^2$, and how does it simplify? The composition $u(v(x))$ simplifies to:

$$u(v(x)) = \sin(x^2)$$

**Error Type:** Not to Prove
**Error Analysis:** The natural language statement presents the content of the formal language as a factual elaboration, rather than treating it as a mathematical proof problem.

Figure 15: A failure case of *Not to Prove*.

**Formal Statement and proof:**

```
import Mathlib
import Aesop
set_option maxHeartbeats 0
open BigOperators Real Nat Topology Rat

/- Let f(x) = x³ - 9x² + 24x be a real-valued function defined on R. Then
   :
   1. f(x) has a local maximum at x = 2.
   2. f(x) has a local minimum at x = 4.
   3. These are the only local extrema of f(x) on R. -/
open Real

noncomputable def f (x : R) : R := x³ - 9 * x² + 24 * x

noncomputable def f_deriv (x : R) : R := 3 * x² - 18 * x + 24

noncomputable def f_second_deriv (x : R) : R := 6 * x - 18

axiom critical_points : f_deriv x = 0 ↔ x = 2 ∨ x = 4

lemma second_derivative_test (x : R) (h : f_deriv x = 0) :
  ((x = 2 ∧ f_second_deriv x < 0) ∨ (x = 4 ∧ f_second_deriv x > 0))
    := by
```

**Transformed Natural Language Statement:**

Consider the real-valued function $f(x) = x^3 - 9x^2 + 24x$ defined on all real numbers. The statement claims that: 1. The function has a local maximum at $x = 2$ 2. The function has a local minimum at $x = 4$ 3. These are the only local extrema of the function on the real numbers

**Error Type:** Not to Prove
**Error Analysis:** The natural language statement expounds on the claim of the formal language, yet it is not a mathematical proof problem itself.

Figure 16: A failure case of *Not to Prove*.

**Formal Statement and proof:**

```
import Mathlib
import Aesop
set_option maxHeartbeats 0
open BigOperators Real Nat Topology Rat

/-- Solve the system:
   x + y + z = a
   x^2 + y^2 + z^2 = b^2
   xy = z^2
   where a and b are constants.
   Give the conditions on a and b so that x, y, z are distinct
       positive numbers. -/
theorem imo_1961_p1 (x y z a b : R)
  (h_0 : 0 < x ∧ 0 < y ∧ 0 < z)
  (h_1 : x ≠ y) (h_2 : y ≠ z) (h_3 : z ≠ x)
  (h_4 : x + y + z = a)
  (h_5 : x^2 + y^2 + z^2 = b^2)
  (h_6 : x * y = z^2) :
  0 < a ∧ b^2 < z^2 ∧ a^2 < 3 * b^2 := by
```

**Transformed Natural Language Statement:**

Given three positive real numbers $x, y, z$ that are all distinct, and two constants $a$ and $b$, solve the system of equations:
1. $x + y + z = a$
2. $x^2 + y^2 + z^2 = b^2$
3. $xy = z^2$
Find the conditions that $a$ and $b$ must satisfy for the system to have distinct positive real solutions $x, y, z$.

**Error Type:** Not to Prove
**Error Analysis:** The natural language statement requires finding the conditions that meet the problem, rather than proving the problem described by the formal language.

Figure 17: A failure case of *Not to Prove*.

**Formal Statement and proof:**

```
import Mathlib
open Real
variable {u v x y : R}
noncomputable def f (u v : R) : R := (u^2 + v^2) * exp (-(u + v))
/-- The function g defined in terms of f -/
noncomputable def g (x y : R) : R := f x (y - x)
axiom f_partial_condition (u v : R) :
  deriv (fun u => f u v) u - deriv (fun v => f u v) v = 2 * (u - v) *
      exp (-(u+v))
axiom f_initial_condition (u : R) :
  f u 0 = u^2 * exp (-u)
axiom partial_g_x (x y : R) :
  deriv (fun x => g x y) x = (4 * x - 2 * y) * exp (-y)
lemma g_expression (x y : R) :
  g x y = (x^2 + (y - x)^2) * exp (-y) := by
  simp only [g, f]
  ring
```

**Transformed Natural Language Proof**:

The proof proceeds as follows:
1. We start with the definition of $g(x, y)$ as $f(x, y - x)$, where $f$ is defined to be $(u^2 + v^2)e^{-(u+v)}$.
2. The goal is to show that $g(x, y)$ equals $(x^2 + (y - x)^2)e^{-y}$.
3. By substituting the definition of $f$ into $g$, we get: $g(x, y) = f(x, y - x) = (x^2 + (y - x)^2)e^{-(x+(y-x))}$
4. Simplifying the exponent: $-(x + (y - x)) = -y$
5. Thus we have $g(x, y) = (x^2 + (y - x)^2)e^{-y}$
6. The proof uses the 'simp' tactic in Lean to simplify the expression, focusing only on the definitions of $g$ and $f$.
7. The 'ring' tactic is then used to verify that the algebraic manipulations are correct.
This shows that the given expression for $g(x, y)$ in terms of $f$ is indeed correct, which supports part (b) of the statement about the form of $f(u, v)$.

**Error Analysis:** This natural language proof is actually an explanation of the formal language proof and cannot exist as an independent solution to the problem. Especially in the key parts of the problem, it only explains the tactics used in the formal language without providing a correct natural language proof.

Figure 18: A failure case of translating formal proof to natural language proof.

**Formal Statement and proof:**

```
import Mathlib
import Aesop
set_option maxHeartbeats 0
open BigOperators Real Nat Topology Rat
/-- If \( Q = 11-5i \), \( E = 11+5i \), and \( D = 2i \), find \( Q\
    cdot E \cdot D \). Show that it is 292i.-/
theorem mathd_algebra_192 (q e d : \mathbb{C})
(h_0 : q = 11 - 5 * Complex.I) (h_1 : e = 11 + 5 * Complex.I)
(h_2 : d = 2 * Complex.I) : q * e * d = 292 * Complex.I := by
rw [h_0, h_1, h_2]
simp only [Complex.ext_iff, Complex.mul_re,
Complex.mul_im, Complex.add_re,
Complex.add_im,
Complex.ofReal_re, Complex.ofReal_im,
Complex.ofReal_add, Complex.ofReal_mul,
Complex.I_re,
Complex.I_im]
norm_num
<;> ring
<;> norm_num
<;> ring
```

**Transformed Natural Language Proof**:

To prove that $Q \times E \times D = 292i$, we follow these steps:
1. First, we substitute the given values into the expression: - Replace $Q$ with $(11 - 5i)$ - Replace $E$ with $(11 + 5i)$ - Replace $D$ with $(2i)$
2. We then expand the multiplication step by step: - First multiply $Q$ and $E$ using complex multiplication rules - Then multiply the result by $D$
3. During the expansion, we: - Separate the real and imaginary parts of the complex numbers - Apply the distributive property of multiplication - Simplify using the fact that $i^2 = -1$ - Combine like terms
4. The simplification process involves: - Calculating the real parts (which should cancel out) - Calculating the imaginary parts (which should combine to 292) - Using basic arithmetic operations (addition, multiplication) - Normalizing the numerical expressions
5. After all simplifications, we verify that: - The real part becomes 0 - The imaginary part becomes exactly 292 - Thus the final result is indeed $292i$

**Error Analysis:** The proof in formal language uses Lean's internal algebraic simplification mechanism, and its correctness can be verified. However, the corresponding natural language proof must provide a detailed simplification process, which the above proof fails to do.

Figure 19: A failure case of ranslating formal proof to natural language proof.

**Formal Statement and proof:**

```
import Mathlib
import Aesop
set_option maxHeartbeats 0
open BigOperators Real Nat Topology Rat

/--  Show that there are infinitely many positive integers m for
    which
      there exists at least one positive integer n such that
        m · n ≤ m + n. -/
theorem amc12a_2002_p6 (n : N) (h_0 : 0 < n) :
  ∃ m, m > n ∧ ∃ p, m * p ≤ m + p := by
  /- For any positive n, choose m = n+1 and p = 1.
     Then (n+1)*1 = n+1 ≤ n+2 = (n+1)+1, so the inequality holds. -/
  use n + 1
  constructor
  · exact Nat.lt_succ_self n         -- n+1 > n
  · use 1
    simp                             -- (n+1)*1 ≤ (n+1)+1
```

**Transformed Natural Language Proof**:

To prove that there are infinitely many positive integers $m$ for which there exists at least one positive integer $n$ such that $m \cdot n \leq m + n$, we proceed as follows:
1. Constructing $m$: For any given positive integer $n$, define $m = n + 1$. This ensures $m > n$ (since $n + 1 > n$).
2. Verifying the inequality: Substitute $m = n + 1$ into $m \cdot n \leq m + n$:

$$(n + 1) \cdot n \leq (n + 1) + n$$

Simplify: $n^2 + n \leq 2n + 1$. For Lean's proof, choosing $p = 1$ simplifies the inequality to:

$$(n + 1) \cdot 1 \leq (n + 1) + 1 \implies n + 1 \leq n + 2$$

which is always true.
3. Conclusion: For every $n$, $m = n + 1$ works with $p = 1$, so there are infinitely many such $m$. The answer is (E) infinitely many.
Note: Lean uses $p = 1$ for simplification, but the key idea of infinitely many $m$ remains.

**Error Analysis:** The formal language proof and the natural language proof adopt different approaches.

Figure 20: A failure case of translating formal proof to natural language proof.

**Evaluation Prompt(RFM Dataset)**:

You are a math teacher highly skilled in evaluating proofs.
**Problem Statement**: [*Original Question*]
**Student's Answer**: [*LLM generated Proof*]

**Your Tasks**
1. **Detailed Analysis**: Read the student's answer and thought process carefully.
- If a claim is obvious in students's answer, you may accept it as valid provided the overall logic is sound.
- If the category **"Others"** is triggered, briefly describe the error in the Analysis section.

2. **Binary Scoring**: After your analysis, output *only* the exact block below, inside triple back-ticks, nothing else:
"
Error Pattern Analysis
- Transformation Error: 1|0
- Over Generalization: 1|0
- Invalid Construction: 1|0
- Wrong Division: 1|0
- Circular Reasoning: 1|0
- Logic Violation: 1|0
- Hidden Assumption: 1|0
- Boundary Neglect: 1|0
- Vague Argument: 1|0
- Vague Argument: 1|0
- Others: 1|0

Overall Correctness
- 1|0
"

**Error Pattern Rubric**
Presented in Figure 22

**Scoring Semantics**
- In **Error Pattern Analysis**: "1" = this error pattern **is present**. "0" = this error pattern **is NOT present**.
- In **Overall Correctness**: "1" = the proof is **completely correct** (no errors). "0" = the proof **contains at least one error**.

**Consistency Rule**
If **any** error pattern is "1", then **Overall Correctness must be "0"**. Only when **all** error patterns are "0" is Overall Correctness "1".

Do not output anything after the code block. Your answer is:

Figure 21: The evaluation prompt we use when assessing the answers to questions in the RFM Dataset generated by various models, which is provided to the *Gemini-2.5-pro* model.

**Error Pattern Rubric**:

1.
Category: **Transformation Error**
Definition: Recasting the target statement into a non-equivalent or strictly weaker one.
Typical example: To prove convergence of $\sum a_n$, only prove $\lim a_n = 0$; or replace "$A \iff B$" with "$A \Rightarrow B$".

2.
Category: **Over Generalization**
Definition: Inferring a universal claim from a few special or hand-picked cases.
Typical example: Verifying for $n = 1, 3, 5$ then claiming the result holds for all $n \in (N)$.

3.
Category: **Invalid Construction**
Definition: Failing to construct an object that should exist, or constructing one that doesn't meet requirements.
Typical example: Claiming a function that is everywhere linear yet nowhere differentiable.

4.
Category: **Wrong Division**
Definition: Partitioning into cases that miss at least one legitimate possibility or overlap.
Typical example: When analyzing the behavior of a function, dividing cases as "always positive," "always zero," and "always negative."

5.
Category: **Circular Reasoning**
Definition: Using the conclusion (or an equivalent reformulation) as a hidden or explicit premise.
Typical example: Assuming $B$ when trying to prove $A \Rightarrow B$.

6.
Category: **Logic Violation**
Definition: A deduction step that contradicts logical or algebraic rules.
Typical example: From $a < b$ and $c < d$ concluding $a - c < b - d$ without checking signs.

7.
Category: **Hidden Assumption**
Definition: Applying a theorem or step whose hypotheses were neither stated nor proven.
Typical example: Differentiating a function known only to be continuous.

8.
Category: **Boundary Neglect**
Definition: Ignoring edge cases, endpoints, or limiting situations so the argument holds only "in the middle."
Typical example: Proving $f(x) = \sqrt{(x)}$ differentiable on $[0, 1]$ without checking at $x = 0$.

9.
Category: **Vague Argument**
Definition: Relying on intuition, diagrams, or "obvious" without formal justification.
Typical example: "The series obviously converges because the terms get smaller."

10.
Category: **Incomplete Proof**
Definition: mitting an essential component such as the converse, base case, or a logical bridge.
Typical example: Proving sufficiency but not necessity in an "if and only if."

11.
Category: **Others**
Definition: Any error not covered by the categories above.

Figure 22: The error pattern rubric used by the prompt shown in Figure 21.

**Lean 4 Evaluation Template w/ Verification**

<TASK_TYPE>
formal_proof_evaluation
<TASK_TYPE>

<THEOREM>
{theorem_statement}
<THEOREM>

<PROOF_ATTEMPT>
{proof_code}
<PROOF_ATTEMPT>

<VERIFICATION_RESULT>
{verification_output}
<VERIFICATION_RESULT>

Please evaluate this Lean 4 proof attempt and provide structured feedback.
Your response must follow this exact format:

<ERROR_ANALYSIS>
[Provide detailed technical analysis of the error, including error type classification, root cause, and severity assessment]
<ERROR_ANALYSIS>

<SCORE>
[Provide a numerical score from 0-100]
<SCORE>

<FEEDBACK>
[Provide specific, actionable suggestions for fixing the proof, including concrete code changes and alternative approaches]
<FEEDBACK>

Figure 23: The prompt with verification provided to evaluator to evaluate formal proof.

**Lean 4 Evaluation Template w/o Verification**

<**TASK_TYPE**>
formal_proof_evaluation
<**TASK_TYPE**>

<**THEOREM**>
{theorem_statement}
<**THEOREM**>

<**PROOF_ATTEMPT**>
{proof_code}
<**PROOF_ATTEMPT**>

Please evaluate this Lean 4 proof attempt and provide structured feedback.
Your response must follow this exact format:

<**ERROR_ANALYSIS**>
[Provide detailed technical analysis of the error, including error type classification, root
cause, and severity assessment]
<**ERROR_ANALYSIS**>

<**SCORE**>
[Provide a numerical score from 0-100]
<**SCORE**>

<**FEEDBACK**>
[Provide specific, actionable suggestions for fixing the proof, including concrete code
changes and alternative approaches]
<**FEEDBACK**>

Figure 24: The prompt without verification provided to evaluator to evaluate formal proof.

**Natural Language Proof Evaluation Prompt w/ Rubrics**

<**TASK_TYPE**>
natural_proof_evaluation
<**TASK_TYPE**>
<**PROBLEM**>
{problem_statement}
<**PROBLEM**>
<**PROOF_ATTEMPT**>
{proof_text}
<**PROOF_ATTEMPT**>
Please evaluate this natural language mathematical proof from the folowing rubrics and provide structured feedback.
<**RUBRICS**>
<Rubrics> <**RUBRICS**>
Your response must follow this exact format:
<**ERROR_ANALYSIS**>
[Follow the above rubrics to provide a detailed conceptual analysis step by step, carefully assessing the proof attempt from each rubric perspective as detailed as possible. You should output a score for each rubric after your analysis, and provide a brief explanation for each score. The scores should be in the range of 0-10 for each rubric, with 0 indicating no evidence of the criterion and 10 indicating perfect adherence to the criterion. For each rubric, please repeat the proof attempt step by step and analyze it according to the rubric.]
<**ERROR_ANALYSIS**>
<**SCORE**>
[In this field, please provide the sum of the scores from all rubrics, which should be a number between 0 and 80, please use addition to calculate the final score step by step and output the final score in the <**SUM**>int<**SUM**> format.
Please first write down the equation for the final score calculation, e.g., "Final Score = int + int + int = int", and then output the final score in the <**SUM**>int<**SUM**> format.]
<**SCORE**>
<**FEEDBACK**>
[In this field, please provide the feedback that can help the student improve their proof attempt. The feedback must be based on the error analysis and scores provided above, and should provide clear guidance. Please do not provide the ground truth of the proof directly.]
<**FEEDBACK**>

Figure 25: The prompt with rubrics provided to evaluator to evaluate natural language proof. The detailed <rubrics> is shown in Figure 26.

**The Rubrics used in Natural Language Proof Evaluation Prompt**

<RUBRICS>
1. Logical Soundness & Step Validity [0-10]
* Content: Whether reasoning is valid, whether logical fallacies or counterexamples exist; whether each reasoning step is correct and reasonable
* Assessment: Check logical derivation relationships, identify fallacy patterns, verify single-step reasoning
2. Completeness of Argument [0-10]
* Content: Whether all necessary cases are covered, whether proof gaps or missing branches exist
* Assessment: Check case coverage, identify unhandled assumptions or boundary conditions
3. Justification & Adequacy [0-10]
* Content: Whether each assertion has sufficient basis, founded on established theorems/axioms/definitions
* Assessment: Verify theoretical support for each key assertion
4. Problem Comprehension & Setup [0-10]
* Content: Whether the problem statement is correctly understood, whether initial assumptions and goals are clear and appropriate
* Assessment: Compare proof setup with problem requirements for consistency
5. Mathematical Rigor [0-10]
* Terminology & Notation: Whether mathematical terms, symbols, definitions, and theorems are used correctly and appropriately
* Computational Accuracy: Whether arithmetic, algebraic, or other mathematical calculation errors are avoided
* Assessment: Verify accuracy of symbolic operations and computational processes
6. Clarity & Presentation [0-10]
* Content: Whether proof structure is logically sound, whether language is clear and comprehensible
* Assessment: Check readability, fluency, and structural organization
7. Relevance & Focus [0-10]
* Content: Whether all statements and calculations are relevant to the proof goal, avoiding redundancy
* Assessment: Identify irrelevant content and reasoning that deviates from the main line
8. Conclusion Completeness [0-10]
* Content: Whether there is a clear concluding statement that summarizes what has been proven
* Assessment: Check correspondence between conclusion and objectives
<RUBRICS>

Figure 26: The rubrics used in prompt shown in Figure 25

**Natural Language Proof Evaluation Prompt**

<**TASK_TYPE**>
natural_proof_evaluation
<**TASK_TYPE**>

<**PROBLEM**>
{problem_statement}
<**PROBLEM**>

<**PROOF_ATTEMPT**>
{proof_text}
<**PROOF_ATTEMPT**>

Please evaluate this natural language mathematical proof and provide structured feedback.
Your response must follow this exact format:

<**ERROR_ANALYSIS**>
[Provide detailed conceptual analysis of mathematical reasoning, including logical structure
assessment, missing components, and error classification]
<**ERROR_ANALYSIS**>

<**SCORE**>
[Provide a numerical score from 0-100]
<**SCORE**>

<**FEEDBACK**>
[Provide educational feedback focusing on mathematical concepts, reasoning gaps, and
suggestions for improvement with clear explanations]
<**FEEDBACK**>

Figure 27: The prompt without rubrics provided to evaluator to evaluate natural language proof.

**Statement and Proof:**

```
import Mathlib
open Real
noncomputable def f (x : R) : R := (1 / cos x) ^2
axiom value_at_pi_over_3 : f (π / 3) = 4
axiom derivative\_of\_f (x : R) : HasDerivAt f (2 * (1 / cos x) ^2 *
    tan x) x
axiom derivative_value_at_pi_over_3 : (2 * (1 / cos (π / 3)) ^2 * tan
    (π / 3)) = 8 * Real.sqrt 3
theorem tangent_line_equation :
∀ x y : R, y = 8 * Real.sqrt 3 * (x - π / 3) + 4 ↔
y - f (π / 3) = (2 * (1 / cos (π / 3)) ^2 * tan (π / 3)) * (x - π /
    3) :=
intro x y
constructor
· intro h
    rw [h, value_at_pi_over_3]
    ring
· intro h
    rw [value_at_pi_over_3] at h
    rw [h]
    ring
```

**Feedback:**

To fix the proof:
1. In the backward direction, after substituting '$f(\pi)/3) = 4$', the equation becomes '$y - 4 = 8\sqrt{3}\left(x - \frac{\pi}{3}\right)$'. Instead of using 'rw [h]', directly add 4 to both sides using 'rw [← h]' or 'have h' : $y = 8\sqrt{3}\left(x - \frac{\pi}{3}\right) + 4 :=$ by rw [h]; ring'.
2. Replace the line 'rw [h]' with 'rw [← h]' to correctly isolate 'y'.
3. Ensure the ring tactic is applied to the transformed equation to simplify it to the target form. Example correction for the backward direction:
· intro h
rw [value_at_pi_over_3] at h
have h' : y = 8 * Real.sqrt 3 * (x - π / 3) + 4 := by rw
[h]; ring
rw [h']

**Comments:**

This feedback is highly beneficial: it correctly identifies the logical error in the proof (the misuse of rw [h] in the backward direction), provides clear and actionable fixes, and includes pedagogical explanations.

Figure 28: A case of *feedback*.

**Statement and Proof:**

```
theorem aime_1990_p4 (x : ℝ) (h₀ : 0 < x) (h₁ : x ^2 - 10 * x - 29 ≠
    0)
    (h₂ : x ^2 - 10 * x - 45 ≠0) (h₃ : x ^2 - 10 * x - 69 ≠0)
    (h₄ : 1 / (x ^2 - 10 * x - 29) + 1 / (x ^2 - 10 * x - 45) - 2 / (
        x ^ 2 - 10 * x - 69) = 0) :
    x = 13 := by
    have x_pos : 0 < x := h₀
    have h₁ : x ^2 - 10 * x - 29 ≠0 := h₁
    have h₂ : x ^2 - 10 * x - 45 ≠0 := h₂
    have h₃ : x ^2 - 10 * x - 69 ≠0 := h₃
    field_simp [h₁, h₂, h₃] at h₄
    linarith
```

**Feedback:**

1. **Missing Quadratic Solution**: After determining y = 39, the proof must solve the quadratic equation x² - 10x - 39 = 0. Add code to derive x = 13 from this equation.
2. **Use of linarith**: linarith is insufficient here because it only solves linear equations. Instead, use algebraic manipulation (e.g., 'simp', 'rw', or 'ring') to solve the quadratic equation.
3. **Correct Algebraic Steps**: Explicitly show the quadratic solution steps, such as completing the square or using the quadratic formula, to justify x = 13.
4. **Example Fix**: Replace 'linarith' with 'have $h_5 : x^2 - 10 * x - 39 = 0 :=$ by rw $[h_y]$', then solve for x using 'simp' or 'ring'.

**Comments:**

This feedback is essential because it precisely diagnoses the critical error—the misuse of linarith for solving quadratic equations—while correctly identifying the core mathematical structure (x² - 10x - 39 = 0) derived from the hypotheses. It mandates explicit solution of the quadratic equation and provides implementable, pedagogically sound steps to complete the proof. By replacing inadequate tactics with algebraic reasoning and leveraging the positivity constraint to eliminate extraneous roots, the proposed fix delivers a complete and verifiable solution.

Figure 29: A case of *feedback*.

2970
2971
2972
2973
2974
2975
2976
2977
2978
2979
2980
2981
2982
2983
2984
2985
2986
2987
2988
2989
2990
2991
2992
2993
2994
2995
2996
2997
2998
2999
3000
3001
3002
3003
3004
3005
3006
3007
3008
3009
3010
3011
3012
3013
3014
3015
3016
3017
3018
3019
3020
3021
3022
3023

**Statement and Proof:**

```
theorem mathd_algebra_480 (f : ℝ → ℝ) (h₀ : ∀ x < 0, f x = -x ^2 -
    1)
  (h₁ : ∀ x, 0 ≤x ∧x < 4 → f x = 2) (h₂ : ∀ x ≥4, f x = Real.sqrt x)
    : f Real.pi = 2 := by
  have h : 0 < Real.pi := Real.pi_pos
  simp_all [h₁]
```

**Feedback:**

1. **Add proof for $\pi < 4$**: Use a theorem like 'Real.pi_lt_four' (if available) to establish 'Real.pi < 4'. If not, construct the proof using real number properties.
2. **Correctly apply $h_1$**: Replace 'simp_all [$h_1$]' with explicit application of $h_1$ using '$h_1$ _ (And.intro (Real.pi_nonneg) (Real.pi_lt_four))' (adjusting theorem names as needed).
3. **Refactor the proof**: Structure the proof as: - Prove '0 ≤ Real.pi' from '0 < Real.pi'. - Prove 'Real.pi < 4'. - Use '$h_1$' with these conditions to conclude 'f Real.pi = 2'.

**Comments:** This feedback is highly beneficial as it precisely identifies the critical gap—the missing proof that $\pi < 4$ required to apply the function's conditional definition—and resolves it by directing the user to the essential Mathlib theorem Real.pi_lt_four. It corrects the ineffective simp_all tactic by prescribing explicit hypothesis application ($h_1$ _ ⟨Real.pi_nonneg, Real.pi_lt_four⟩), transforming an incomplete automation attempt into a robust, verifiable proof that adheres strictly to the function's domain conditions while enhancing pedagogical clarity.

Figure 30: A case of *feedback*.

**Statement and Proof:**

Given positive integers n and k, there is a finite set $A_{(n,k)} = \{\alpha_1, \alpha_2, ..., \alpha_n\}$ where $\alpha_i = (a_{i,1}, a_{i,2}, ..., a_{i,(i-1)k+1})$, $a_{i,1}, a_{i,2}, ..., a_{i,(i-1)k+1} \in \{0, 1\}$ and $a_{i,1} + a_{i,2} + ... + a_{i,(i-1)k+1} = 1$ for $i = 1, 2, ..., n$. For an n-term sequence $\Omega : (1, x_1), (2, x_2), ..., (n, x_n)$, if $x_1 = 1$ and $x_{i+1} - x_i \in \{0, 1, ..., k\}$ for $i = 1, 2, ..., n - 1$, then it is called a "G-sequence". At this time, denote $G(\Omega) = a_{1,x_1} + a_{2,x_2} + ... + a_{n,x_n}$. Prove that for any set $A_{(n,k)}$, there exists a "G-sequence" $\Omega$ such that $(k + 1)^{G(\Omega)} > kn$.

**Proof:**

To prove that for any set $A_{(n,k)}$, there exists a G-sequence $\Omega$ such that $(k + 1)^{G(\Omega)} > kn$, we can use the probabilistic method and properties of G-sequences.

Key Observations: 1. **G-sequences**: A G-sequence $\Omega$ starts at $x_1 = 1$ and each subsequent $x_{i+1}$ satisfies $0 \le x_{i+1} - x_i \le k$. There are $(k + 1)^{n-1}$ such sequences since each step has $k + 1$ choices.

2. **Function $G(\Omega)$**: $G(\Omega)$ counts the number of positions $i$ where $x_i$ is the position of the 1 in $\alpha_i$. Let $t_i$ denote the position of the 1 in $\alpha_i$; then $G(\Omega) = \sum_{i=1}^{n} \mathbf{1}_{\{x_i = t_i\}}$.

3. **Probabilistic Method**: Consider a random G-sequence $\Omega$ where each step is chosen uniformly. Define $X = G(\Omega)$. We need to show there exists a sequence with $(k + 1)^X > kn$.

Analysis: - The total number of G-sequences is $(k + 1)^{n-1}$. - For each sequence, $(k + 1)^{G(\Omega)}$ is the product of $(k + 1)$ for each $i$ where $x_i = t_i$, and 1 otherwise.

Sum of $(k + 1)^{G(\Omega)}$: The sum of $(k + 1)^{G(\Omega)}$ over all G-sequences is at least $(k + 1)^n$. This is because each sequence contributes at least $k + 1$ (since $x_1 = t_1 = 1$ always, so $G(\Omega) \ge 1$), and there are $(k + 1)^{n-1}$ sequences.

Pigeonhole Principle: If all sequences had $(k + 1)^{G(\Omega)} \le kn$, then the total sum would be at most $kn \times (k + 1)^{n-1}$. However, we know the total sum is at least $(k + 1)^n$, so:

$$(k + 1)^n > kn \times (k + 1)^{n-1}$$

Simplifying, this implies $k + 1 > kn$, which is true for $n = 1$. For $n \ge 2$, the exponential growth of $(k+1)^n$ compared to the linear growth of $kn$ ensures that there must exist a sequence where $(k + 1)^{G(\Omega)} > kn$.

**Feedback:**

The proof attempt demonstrates an understanding of the probabilistic method and the pigeonhole principle but contains critical errors in the application of these concepts. Here's a breakdown of the feedback:
1. **Conceptual Understanding**: The user correctly identifies the total number of G-sequences and the structure of $G(\Omega)$. They also attempt to use the probabilistic method and the pigeonhole principle, which are valid strategies for such problems.
2. **Logical Gaps**: The core error lies in the incorrect inequality $(k + 1)^n > kn \cdot (k + 1)^{n-1}$. This inequality is not generally true, and the proof fails to address cases where it does not hold (e.g., $n = 2, k = 1$). The argument also misapplies the pigeonhole principle by assuming that the average value being greater than $kn$ guarantees the existence of a sequence exceeding $kn$.
3. **Suggestions for Improvement**: - **Re-examine the Inequality**: The proof should address the general case by considering the relationship between $k + 1$ and $kn$ for all $n$ and $k$. For example, when $k + 1 \le kn$, alternative methods (e.g., explicit construction of a G-sequence) may be necessary. - **Clarify the Pigeonhole Principle Application**: The argument should explicitly show that the average value of $(k + 1)^{G(\Omega)}$ being at least $k + 1$ ensures the existence of a sequence with $(k + 1)^{G(\Omega)} > kn$, which requires a more rigorous analysis of the total sum and individual contributions. - **Consider Edge Cases**: The proof should test edge cases like $n = 1$ or $k = 1$ to ensure the result holds universally.

**Comments:** The feedback is highly beneficial as it specifically identifies critical errors in the proof, such as the incorrect application of the inequality $(k+1)^n > kn \cdot (k+1)^{n-1}$ and its failure in cases like $n = 2, k = 1$, while constructively suggesting improvements like re-examining the inequality for all $n$ and $k$, clarifying the pigeonhole principle usage, and testing edge cases; this precise, actionable guidance helps the author address logical gaps and enhance the proof's rigor, making it both educational and effective for refinement.

Figure 31: A case of *feedback*.

