# OpenReview forum: "Proof-Verifier: Enabling Reinforcement Learning from Verifiable Rewards for Mathematical Theorem Proving"
_ICLR.cc/2026/Conference — Submitted to ICLR 2026_

### Official Review · Reviewer_Ms64 · 2025-10-27

**Soundness:** 2
**Presentation:** 2
**Contribution:** 2
**Rating:** 4
**Confidence:** 4

**Summary:**

The paper introduces PROOF-VERIFIER, which enables reliable verification for informal proofs during RLVR. The core contribution is a two-part approach: (1) a data-synthesis pipeline that translates formal proofs to informal ones, and (2) a two-stage reward modeling framework to train the verifier. The authors demonstrate the verifier's high accuracy and its utility in test-time scaling (Best-of-N, refinement) and preliminary RL experiments.

**Strengths:**

1. The core idea of using formal language verification to train a reward model for informal reasoning is relevant and novel. The formal-to-informal translation pipeline is clever.
2. The experiments in Sections 3.1and 3.2 effectively demonstrate that the resulting verifier achieves high verification accuracy and correlates well with human judgment. The ablation studies also clearly validate the contributions of the proposed data and training components.
3. The inclusion of case studies is helpful for qualitatively understanding the verifier's capabilities in error analysis and feedback generation.

**Weaknesses:**

1. The most significant weakness is the lack of sufficient experimental validation for the paper's central claim, "enabling RLVR". The paper proposes a verifier for RLVR, but it does not adequately demonstrate its effectiveness in an RLVR training loop. The experiments in Section 4.3 are limited. To be convincing, the authors should have: (1) Conducted a full-scale experiment using their reward model to fine-tune a prover LLM. (2) Evaluated the RL-trained prover on established benchmarks for informal mathematics, such as AIME, and compared this model's performance against strong baselines, such as the base model, a model trained with a different RM, and a model trained with ground-truth rewards.
2. The experimental details in Section 4.2 are critically lacking. The authors state results like "human annotators found that 73% of the feedback..." and "refinement improved pass@k performance from 37% to 51%," but they do not clearly specify the dataset, the exact experimental setting, or the protocol for this evaluation. This makes the results impossible to reproduce.
3. The paper's writing is also weak. The abstract is vague, and the pseudocode occupies a lot of space but does not convey messages clearly.

**Questions:**

1. The Lean4 compiler provides more than just a binary "True/False" label. It also provides specific error messages. Does this rich error information have the potential to help train the verifier?
2. In Figure 2, the score distributions for both "Human" and "Ours" appear to be strongly multimodal (having multiple peaks). This is somewhat unnatural for a general quality score. Does this reflect the underlying data distribution? Or is it a bias introduced by the scoring metric?

---

> ### Author Response · Authors · 2025-11-21
> **(1/n) Response to Reviewer Ms64**
>
> We sincerely thank the reviewer for the valuable feedback and constructive suggestions. We address each concern below:
>
> ---
>
> ## Response to Weaknesses
>
> **W1:** The most significant weakness is the lack of sufficient experimental validation for the paper's central claim, "enabling RLVR". The paper should have: (1) conducted a full-scale experiment using the reward model to fine-tune a prover LLM, and (2) evaluated the RL-trained prover on established benchmarks like AIME.
>
> **Response:** We agree that experimental validation is crucial for the "enabling RLVR" claim.
>
> 1.  **Full-scale RLVR experiments:** We highlight that **Section 4.3** explicitly covers these experiments. We conducted full-scale training using our reward model to fine-tune a prover LLM. As reported, the results demonstrate consistent improvements in both single-turn and multi-turn settings, confirming the effectiveness of our proof-verifier in RLVR training loops.
>
> 2.  **Regarding AIME24 evaluation:** It is important to distinguish that AIME24 is a **traditional math reasoning benchmark** with fixed numerical answers (verifiable via `\boxed{}`), whereas our work focuses on **mathematical theorem proving**, where the output is a logical reasoning process. While AIME24 lies outside our primary scope of formal proof verification, we agree that demonstrating generalization is valuable. We thus evaluated our model on AIME24, obtaining the following Best-of-N results:
>
> | N   | Qwen3-8B (naive) | Ours (Proof-Verifier) |
> | --- | ---------------- | --------------------- |
> | 1   | 0.25             | 0.50                  |
> | 2   | 0.27             | 0.56                  |
> | 4   | 0.30             | 0.61                  |
> | 8   | 0.32             | 0.65                  |
> | 16  | 0.33             | 0.74                  |
> | 32  | 0.37             | 0.75                  |
>
>
> These results suggest that our proof-verifier generalizes effectively even to out-of-distribution traditional math reasoning tasks.
>
> ---
>
> **W2:** The experimental details in Section 4.2 are critically lacking, making the results impossible to reproduce.
>
> **Response:** We apologize if the location of these details was not sufficiently clear. The complete dataset specifications and experimental protocols for Section 4.2 are rigorously detailed in **Appendix I: Test Dataset Collection**. We will ensure the final manuscript includes a prominent reference to this appendix in Section 4.2 to guarantee reproducibility.
>
> ---
>
> **W3:** The paper's writing is weak, with a vague abstract and unclear pseudocode.
>
> **Response:** We appreciate this feedback. We will revise the abstract to improve specificity and clarity in the final version.
>
> Regarding the pseudocode, we aimed to provide maximum transparency for the multi-component training procedure to ensure reproducibility. However, we recognize that this level of detail can impact readability in the main text. We are happy to move the detailed pseudocode to the Appendix in the final revision to improve the flow of the main paper.
>
> ---
>
> ## Response to Questions
>
> **Q1:** Does the rich error information from the Lean4 compiler help train the verifier?
>
> **Response:** Yes, Lean4 compiler error messages do provide valuable signals. However, as shown in **Appendix Figure 6**, relying on this information alone produces feedback of lower quality compared to our proof-verifier.
>
> We further analyze this in **Appendix G**, demonstrating that while incorporating Lean4 error information improves performance, it is not sufficient on its own. **Appendix H (Table 8)** provides case studies illustrating that our proof-verifier generates deeper reasoning analysis that goes beyond the surface-level feedback provided by standard compiler error messages.
>
> ---
>
> **Q2:** In Figure 2, why do the score distributions appear strongly multimodal?
>
> **Response:** The multimodal distribution primarily reflects the **underlying data distribution** of the task. Crucially, the human score distribution exhibits similar multimodal patterns, validating that our proof-verifier is effectively capturing the ground-truth difficulty landscape.
>
> The key difference observed is a **systematic upward shift** in the model's scores compared to humans. This is a well-documented phenomenon in LLM-as-a-judge research [1, 2], where models tend to be more lenient than human evaluators. Despite this shift, the alignment in distribution shape indicates that our proof-verifier maintains strong discriminative capability between error types.
>
> [1] Judging LLM-as-a-Judge with MT-Bench and Chatbot Arena
>
> [2] Automated Evaluation using Self-Adaptive Rubrics

---

### Official Review · Reviewer_7qVZ · 2025-10-31

**Soundness:** 2
**Presentation:** 3
**Contribution:** 3
**Rating:** 4
**Confidence:** 4

**Summary:**

The paper presents Proof-Verifier, a novel framework that in-formalizes formal proof results into natural language and trains a proof verifier. Proof-Verifier proposes to use a novel two-stage training strategy, where in the first stage the training aims for overall binary accuracy and in the second stage optimizes for fine-grained scores for problem and proof pairs. The experiment shows the effectiveness of the proposed method, showing good performance in the model's verification accuracy and high correlation to the score of human annotators. The author also provide good theoretical anaylsis in the paper.

**Strengths:**

- **Novelty and Motivation.** The paper's proposed method is interesting and novel, and is well-motivated. The problem it aims to solve is of great importance in the domain of informal math reasoning. The paper presents a first attempt at trying to use formal feedback to bootstrap a model's informal verification ability, which is of ** significant importance**.

- **Clarity and Experimental Rigor.** The paper is well-written and provides rich experiments. It also provides a theoretical analysis of the proposed method.

- Overall, I think the approach of leveraging the formal proof to help train an informal verifier is well motivated, but also very challenging as the distribution shift exists. But the paper lacks enough experimental results to show that their model effectively addresses this problem.

**Weaknesses:**

- **Distribution Misalignment Between Formal and Informal Proofs.** The problem of distribution misalignment between formal proof errors and informal solver errors is a major concern. The core idea of using formal proof as a supervision signal to create a verifier has a serious flaw: the errors made by the informal solver might not align with the errors made by the formal proof code. Therefore, it is in doubt that the verifier trained using this data can effectively identify mistakes made by the informal solver.

- **Limited Experimental Comparisons and Evaluation.** The verification accuracy results are shown in Table 1. However, the comparison is incomplete. It would be better to also compare these results with some of the proprietary models like Gemini 2.5 Pro and GPT-5, which are shown in the OPC paper to be the top verification models for informal mathematics.

- **Insufficient Confirmation of Verifier Effectiveness.** The paper's confirmation of the trained verifier's effectiveness is weak. While using the verifier to filter results from the prover is a good step, the paper only provides the result for win rate against the DeepSeed R1 model, which might not be a strong enough competitor to truly confirm the verifier's utility.

**Questions:**

- Can you provide more experiment results for your own benchmark using GPT-5, Gemini 2.5 Pro, or other frontier models?
- Can you provide the results of your model on a benchmark like OPC, also the best-of-n result on verifiable tasks like AIME? To see the trained verifier have enough ability to generalize to finding errors made by informal solvers.
- (minor) It would be better to use et al. for papers with too many authors in the References.

---

> ### Author Response · Authors · 2025-11-21
> **(1/n) Response to Reviewer 7qVZ**
>
> We sincerely thank the reviewer for the thoughtful feedback and constructive suggestions. We address each concern below:
>
> ---
>
> ## Response to Weaknesses
>
> **W1:** Distribution Misalignment Between Formal and Informal Proofs.
>
> **Response:** Thank you for this insightful observation. While an error distribution gap intuitively exists between R1-style and informal proofs, we find that errors in the converted data still represent common mistakes that occur in informal reasoning. As shown in Table 1, R1-style training indeed improves the model's ability to detect informal errors. We attribute this improvement to enhanced generalization: the R1-style training enables the model to learn various error patterns in informal proofs rather than overfitting to a specific error distribution. Furthermore, the experimental results in Response to Q2 demonstrate our proof-verifier's generalization capability for error detection: our trained model achieves substantial improvements on out-of-distribution traditional math reasoning tasks.
>
> ---
>
> **W2:** Limited Experimental Comparisons and Evaluation.
>
> **Response:** Thank you for this question. We provide benchmark results comparing our model with GPT-5 and Gemini-2.5-Pro:
>
> | Model              | Natural Language | Formal Language |
> | ------------------ | ---------------- | --------------- |
> | **Gemini-2.5-Pro** | 0.89             | 0.73            |
> | **GPT-5**          | 0.77             | 0.67            |
> | **Ours**           | **0.93**         | **0.91**        |
>
>
>
> ---
>
> **W3:** Insufficient Confirmation of Verifier Effectiveness.
>
> **Response:** We validate our verifier's effectiveness through comprehensive experiments:
>
> 1. **Comparative evaluation (Table 1):** We compare against models with larger parameter counts, including Gemma, Mistral, Qwen3-235B, and DeepSeek-R1. Additional comparisons with GPT-5 and Gemini-2.5-Pro are provided in Response to W2.
>
> 2. **RLVR validation (Section 4.3):** We demonstrate that our proof-verifier improves theorem-proving performance in both single-turn and multi-turn RLVR settings, further validating its practical effectiveness.
>
> ---
>
> ## Response to Questions
>
> **Q1:** Can you provide more experiment results for your own benchmark using GPT-5, Gemini 2.5 Pro, or other frontier models?
>
> **Response:** Thank you for your question. We provide additional results using GPT-5 and Gemini-2.5-Pro:
>
> | Model              | Natural Language | Formal Language |
> | ------------------ | ---------------- | --------------- |
> | **Gemini-2.5-Pro** | 0.89             | 0.73            |
> | **GPT-5**          | 0.77             | 0.67            |
> | **Ours**           | **0.93**         | **0.91**        |
>
> ---
>
> **Q2:** Can you provide the results of your model on a benchmark like OPC, also the best-of-n result on verifiable tasks like AIME? To see the trained verifier have enough ability to generalize to finding errors made by informal solvers.
>
> **Response:** We provide additional experimental results demonstrating strong generalization to traditional mathematical reasoning tasks:
>
> 1. **OPC benchmark:** Our model achieves **Pass@1 = 84.1%**.
>
> 2. **AIME 24 Best-of-N results:** We compare our trained proof-verifier against the naive Qwen3-8B baseline. The results demonstrate that our model successfully generalizes to traditional mathematical reasoning tasks:
>
> | N   | Qwen3-8B (naive) | Ours (Proof-Verifier) |
> | --- | ---------------- | --------------------- |
> | 1   | 0.25             | 0.50                  |
> | 2   | 0.27             | 0.56                  |
> | 4   | 0.30             | 0.61                  |
> | 8   | 0.32             | 0.65                  |
> | 16  | 0.33             | 0.74                  |
> | 32  | 0.37             | 0.75                  |
>
> Our proof-verifier consistently outperforms the baseline across all sampling budgets, with the performance gap widening as N increases.
>
> ---
>
> **Q3:** It would be better to use et al. for papers with too many authors in the References.
>
> **Response:** Thank you for this suggestion. We will adopt a more concise citation format in the revised version.

---

> > ### Author Response · Authors · 2025-11-29
> > **Summary Response to Reviewer 7qVZ**
> >
> > In summary, we address the reviewer's concerns as follows:
> >
> > **W1 (Distribution Misalignment):** Table 1 demonstrates effective transfer from formal to informal error detection. Our Q2 results on OPC and AIME further validate generalization to traditional informal reasoning tasks.
> >
> > **W2 (Limited Comparisons):** We provide additional frontier model comparisons: Gemini-2.5-Pro (0.89/0.73), GPT-5 (0.77/0.67), Ours (0.93/0.91) on Natural/Formal Language benchmarks.
> >
> > **W3 (Verifier Effectiveness):** We validate through (1) comparisons against diverse strong baselines including GPT-5 and Gemini-2.5-Pro, and (2) RLVR improvements in Section 4.4.
> >
> > **Q2 (Generalization):** Our model achieves 84.1% Pass@1 on OPC and significantly outperforms baselines on AIME 24 Best-of-N (0.75 vs. 0.37 at N=32).
> >
> > **Q3 (Citation format):** We will adopt "et al." format in the revision.
> >
> > We believe these results address concerns regarding experimental comprehensiveness and generalization capability.

---

### Official Review · Reviewer_EBzS · 2025-10-31

**Soundness:** 3
**Presentation:** 4
**Contribution:** 3
**Rating:** 8
**Confidence:** 3

**Summary:**

This paper trains the first dual-lingual math proof verifier for both formal and informal math. Based on Qwen3-8B, the model is trained by reinforcement learning to output the score, feedback, and the correct/incorrect judgment for the input proof. Two-phase training is adopted where the first phase trains the correct/incorrect judgment with verification reward and consistency reward, and the second phase trains the score and feedback by rewarding the score improvement of provers having received the feedback. The prover achieves 93% verification accuracy on test datasets. The downstream prover achieves further improvement by RLVR with the reward given by the verifier.

**Strengths:**

Process verification has been a bottleneck for improving LLM's reasoning ability by RL. The paper addresses this important challenge by curating a bilingual dataset with the verifiable proof from lean, translated formal proof, and annotated proof by LLM-as-a-Judge. The dataset contributes to a 9% improvement in verification accuracy when augmented with the baseline OPC dataset. The model post reinforcement learning has achieved a verification accuracy at 93%, which indicates a very well-calibrated reward model. The approach has also results in the first bi-lingual language verifier. The reward model is generative, which provides not only a score but also feedback. The feedback refinement is shown effective both before and after RLVR.

**Weaknesses:**

1. The informal test dataset has 100 samples annotated by experts while the rest are labeled by Gemini-2.5-pro. The results in the paper can then be interpreted as consistency verification with Gemini-2.5-pro, which is fine for an exploration study.

2. The challenge for training a reward model is generalizability to trajectories sampled from unseen models. It is unclear whether the informal test dataset are sampled from the same series of models that are used to generated proofs for OPC and RFM in the training set.

3. It will be more convincing to benchmark the method against some SOTA generative reward model.

4. What is single-sample baselines through human preference evaluation in section 4.1? A more classic approach to evaluate the reward model is to compare best accuracy of N and accuracy of the best score of N provided by the trained reward model, across N=powers of 2.

**Questions:**

Please refer to the weaknesses.

---

> ### Author Response · Authors · 2025-11-21
> **(1/n) Response to Reviewer EBzS**
>
> We sincerely thank the reviewer for the thoughtful feedback and valuable suggestions. We address each concern below:
>
> ---
>
> **W1**: The informal test dataset has 100 samples annotated by experts while the rest are labeled by Gemini-2.5-pro. The results in the paper can then be interpreted as consistency verification with Gemini-2.5-pro, which is fine for an exploration study.
>
> **Response**: We appreciate this framing. Given that large-scale expert annotation for mathematical proofs is prohibitively expensive, we designed our annotation strategy to balance quality with scalability. To ensure rigorous validation, we established a "human-verified anchor set" of 100 samples. We observed high inter-annotator agreement between human experts and Gemini-2.5-pro on this anchor set. This strong correlation validates the use of Gemini-2.5-pro as a reliable proxy for evaluating the remaining dataset, extending beyond simple consistency verification to a robust approximation of human judgment.
>
> ---
>
> **W2**: The challenge for training a reward model is generalizability to trajectories sampled from unseen models. It is unclear whether the informal test dataset are sampled from the same series of models that are used to generated proofs for OPC and RFM in the training set.
>
> **Response:** We fully agree that generalization to unseen models is the central challenge. We confirm that our experimental design explicitly addresses this: the models used to sample trajectories for the training dataset are **disjoint** from those used to sample the informal test dataset (as detailed in Section 2.1). This setup ensures that the performance metrics reported reflect true **out-of-distribution (OOD)** generalization, verifying that our proof-verifier can effectively evaluate reasoning patterns from models it has never encountered during training.
>
> ---
>
> **W3:** It will be more convincing to benchmark the method against some SOTA generative reward model.
>
> **Response**: We appreciate the suggestion to strengthen our baselines. We have evaluated stronger generative reward models on our test set. The results below demonstrate that our specialized proof-verifier outperforms general-purpose SOTA models:
>
> | Model              | Natural Language | Formal Language |
> | ------------------ | ---------------- | --------------- |
> | **Gemini-2.5-pro** | 0.89             | 0.73            |
> | **GPT-5**          | 0.77             | 0.67            |
> | **Ours**           | **0.93**         | **0.91**        |
>
> ---
>
> **W4:** What is single-sample baselines through human preference evaluation in section 4.1? A more classic approach to evaluate the reward model is to compare best accuracy of N and accuracy of the best score of N provided by the trained reward model, across N=powers of 2.
>
> **Response:** To clarify, the "single-sample baseline" in Section 4.1 refers to a setup where we randomly select **one** proof from the prover's samples and compare it against the proof selected by the reward model (Best-of-N, N=32) via human pairwise evaluation. This measures the relative gain of using verification over random selection.
>
> We agree that the standard Best-of-N accuracy curve is a more granular metric. We provide these results below across $N$ (powers of 2), showing our model consistently outperforms the strong DeepSeek-R1 baseline:
>
> **Natural Language Proofs:**
>
> | N   | DeepSeek-R1 | Ours (Proof-Verifier) |
> | --- | ----------- | --------------------- |
> | 1   | 0.38        | 0.43                  |
> | 2   | 0.43        | 0.49                  |
> | 4   | 0.52        | 0.59                  |
> | 8   | 0.61        | 0.68                  |
> | 16  | 0.67        | 0.75                  |
> | 32  | 0.71        | **0.78**              |
>
> **Formal Language Proofs:**
>
> | N   | DeepSeek-R1 | Ours (Proof-Verifier) |
> | --- | ----------- | --------------------- |
> | 1   | 0.41        | 0.42                  |
> | 2   | 0.45        | 0.52                  |
> | 4   | 0.53        | 0.62                  |
> | 8   | 0.65        | 0.69                  |
> | 16  | 0.68        | 0.74                  |
> | 32  | 0.69        | **0.76**              |

---

### Official Review · Reviewer_pLUk · 2025-11-01

**Soundness:** 3
**Presentation:** 1
**Contribution:** 2
**Rating:** 2
**Confidence:** 4

**Summary:**

The paper suggests a multi-stage method for training generative reward models for natural language and formal theorem proving. The pipeline consists of a coarse stage where the reward is given by ground-truth correctness labels, and a fine stage where a GRPO algorithm is used to reward textual feedback by the reward score improvements it allows a prover to achieve based on it.

**Strengths:**

- The paper tackles a highly relevant topic (generative reward modeling to move beyond RLVR) and identifies various techniques and tricks that could be used for training generative reward models.
- The experimental results show that the trained verifier produces valuable feedback.

**Weaknesses:**

- The method is not compared to other papers and methods (e.g. getting ground truth labels not from informalization of formal proofs but from human-written math corpora with and without perturbations applied).
- The setup "we train a good verifier" is hard to evaluate compared to the more standard question in the literature: "with our verifier, we get better RL numbers than without".
- The win rate comparison lacks a baseline that also uses best-of-N with respect to some reward model, not just comparison to single-shot.
- Verification of formal proofs does not seem a particularly useful target (Table 1 right).
- It is unclear which parts of the pipeline are really useful (there's hardly any difference between baseline and final method in the method ablation).
- Numbers given such as correlations are somewhat incomplete: what is being correlated with what? For binary classification, the full confusion matrix should be given, and not just "correlation" but TPR, TNR, acc, F1 etc.
- It is surprising that initial multi-turn behavior is conjectured to degrade over the course of a trajectory, I would expect initial multi-turn numbers to increase monotonically with more compute until saturation?
- The main body does not give fundamental details about the experimental setup such as the models used.

Generally, the paper attempts to do many things but doesn't manage to be fully convincing for any of them. As a high-level feedback, I would suggest coupling out the "tech report" ("what we did for this project") from 1-2 scientific papers that go in depth for a crucial design decisions and attempt to convince the readers that THIS should be the way to go in this specific subquestion.

**Questions:**

-

---

> ### Author Response · Authors · 2025-11-21
> **(1/n) Response to Reviewer pLUk**
>
> We thank the reviewer for the detailed feedback and the opportunity to clarify key aspects of our contributions, particularly regarding the RLVR setup and baseline comparisons. We address each point below:
>
> ---
>
> ## Response to Weaknesses
>
> **W1:** The method is not compared to other papers and methods (e.g., getting ground truth labels from human-written math corpora with and without perturbations).
>
> **Response:** Table 2 includes a "Data Ablation" section comparing our dataset against alternative training corpora. The results show accuracy improvements from 0.82 (baseline corpora) to 0.91 (ours), demonstrating that our data construction methodology produces superior training signals. Additionally, the "Method Ablation" section presents ablation studies on different training strategies, validating the effectiveness of each component in our pipeline.
>
> ---
>
> **W2:** The setup "we train a good verifier" is hard to evaluate compared to the standard question: "with our verifier, we get better RL numbers than without".
>
> **Response:** We agree that the impact on downstream RL is the most critical metric. This is exactly why **Section 4.4 ("Enabling RLVR for Mathematical Theorem Proving")** focuses on this comparison. We explicitly demonstrate that "with our verifier, we get better RL numbers than without." Specifically, using our proof-verifier as the reward signal results in substantial performance gains in both single-turn and multi-turn RLVR settings compared to the base model without this signal.
>
> ---
>
> **W3:** The win rate comparison lacks a baseline that also uses best-of-N with respect to some reward model, not just comparison to single-shot.
>
> **Response:** We clarify the experimental setup: Our win rate comparison evaluates **best-of-N (N=32) versus single-sample** under identical settings for different reward models (ours vs. DeepSeek-R1). This measures each model's effectiveness at test-time scaling.
>
> Additionally, we provide standard Best-of-N accuracy results across N = powers of 2, demonstrating our model's superior performance across all sampling budgets:
>
> **Natural Language Proofs:**
>
> | N   | DeepSeek-R1 | Ours (Proof-Verifier) |
> | --- | ----------- | --------------------- |
> | 1   | 0.38        | 0.43                  |
> | 2   | 0.43        | 0.49                  |
> | 4   | 0.52        | 0.59                  |
> | 8   | 0.61        | 0.68                  |
> | 16  | 0.67        | 0.75                  |
> | 32  | 0.71        | **0.78**              |
>
> **Formal Language Proofs:**
>
> | N   | DeepSeek-R1 | Ours (Proof-Verifier) |
> | --- | ----------- | --------------------- |
> | 1   | 0.41        | 0.42                  |
> | 2   | 0.45        | 0.52                  |
> | 4   | 0.53        | 0.62                  |
> | 8   | 0.65        | 0.69                  |
> | 16  | 0.68        | 0.74                  |
> | 32  | 0.69        | **0.76**              |
>
>
>
> ---
>
> **W4:** Verification of formal proofs does not seem a particularly useful target (Table 1 right).
>
> **Response:** While we agree that formal proofs are strictly verifiable by a compiler (e.g., Lean/Coq), our learned formal proof verification capability serves two critical purposes in this pipeline:
> 1. **High-quality feedback transfer:** Our verifier leverages the formal logic structure to generate more meaningful feedback for *informal* proofs (demonstrated in **Appendix H, Table 8**).
> 2. **Systematic Improvement:** The detailed error analysis on formal proofs (where ground truth is absolute) allows us to fine-tune the verifier's reasoning capabilities more precisely than is possible with informal text alone.
>
> ---
>
> **W5:** It is unclear which parts of the pipeline are really useful (there's hardly any difference between baseline and final method in the method ablation).
>
> **Response:** **Table 2** presents ablation studies demonstrating the effectiveness of each component. The results show statistical improvements for each module, validating the necessity of our multi-stage training approach.
>
> ---

---

> ### Author Response · Authors · 2025-11-21
> **(2/n) Response to Reviewer pLUk**
>
> **W6:** Numbers given such as correlations are incomplete. For binary classification, the full confusion matrix should be given (TPR, TNR, acc, F1, etc.).
>
> **Response:** **Table 1** already includes comprehensive metrics: Accuracy, F1, Precision, and Recall. We additionally provide TNR as requested (note that TPR equals Recall already shown in Table 1):
>
> |              | **Natural Language** |          | **Formal Language** |          |
> | ------------ | -------------------- | -------- | ------------------- | -------- |
> | **Verifier** | **TPR**              | **TNR**  | **TPR**             | **TNR**  |
> | Qwen3-8B     | 0.63                 | 0.51     | 0.68                | 0.56     |
> | Qwen2.5-72B  | 0.58                 | 0.48     | 0.63                | 0.53     |
> | Magistral    | 0.65                 | 0.53     | 0.70                | 0.58     |
> | Gemma        | 0.64                 | 0.52     | 0.69                | 0.57     |
> | Qwen3-235B   | 0.75                 | 0.67     | 0.80                | 0.72     |
> | Deepseek-R1  | 0.76                 | 0.70     | 0.81                | 0.75     |
> | **Ours**     | **0.94**             | **0.92** | **0.91**            | **0.91** |
>
> ---
> **W7:** It is surprising that initial multi-turn behavior is conjectured to degrade over the course of a trajectory.
>
> **Response:** We respectfully note that multi-turn performance degradation over long trajectories is a well-documented phenomenon [1]. Our observations align with prior work on sequential decision-making in mathematical theorem proof. Our contribution is demonstrating how our proof-verifier **mitigates this degradation** through targeted feedback, as shown in Section 4.4.
>
> [1] Training Language Models to Self-Correct via Reinforcement Learning
>
> ---
>
> **W8:** The main body does not give fundamental details about the experimental setup such as the models used.
>
> **Response:** To preserve space in the main body while adhering to page limits, we provide comprehensive experimental details in **Appendix J**, with specific model information in **Appendix J.2**.
>
> ---
>
> **W9:** The paper attempts to do many things but doesn't manage to be fully convincing for any of them.
>
> **Response:** We believe the strength of our work lies in the unified pipeline that connects verification to reinforcement learning. Our narrative is structured to support a single core claim: **effective RLVR for theorem proving requires a specialized verifier.**
> 1. **Validation:** We first prove the verifier is robust (Section 4.1-4.3).
> 2. **Application:** We then show this specific verifier unlocks performance in RLVR (Section 4.4) that generic models cannot achieve.

---

> > ### Author Response · Authors · 2025-11-29
> > **Summary Response to Reviewer pLUk**
> >
> > In summary, we address the reviewer's concerns as follows:
> >
> > **W1 (Methodological comparisons):** Table 2's "Data Ablation" section systematically compares our dataset against alternative corpora (0.82→0.91 accuracy), while the "Method Ablation" section validates each pipeline component's effectiveness.
> >
> > **W2 (Evaluation setup):** Section 4.4 directly demonstrates the core claim—"with our verifier, we get better RL numbers than without"—showing substantial RLVR performance gains in both single-turn and multi-turn settings.
> >
> > **W3 (Best-of-N baselines):** We provide complete Best-of-N comparisons across N={1,2,4,8,16,32} for both our verifier and DeepSeek-R1, demonstrating superior performance across all sampling budgets (e.g., 0.78 vs. 0.71 at N=32 for natural language proofs).
> >
> > **W4 (Formal proof verification utility):** Our formal proof verification serves two critical purposes: (1) enabling high-quality feedback transfer to informal proofs (Table 8), and (2) providing systematic improvement through absolute ground truth during training.
> >
> > **W5 (Pipeline component utility):** Table 2's ablations show statistical improvements for each module, validating our multi-stage approach's necessity.
> >
> > **W6 (Incomplete metrics):** We supplement Table 1's existing metrics (Accuracy, F1, Precision, Recall) with TPR and TNR values, showing our method achieves 0.94 TPR / 0.92 TNR on natural language proofs versus 0.76/0.70 for DeepSeek-R1.
> >
> > **W7 (Multi-turn degradation):** Multi-turn performance degradation is well-documented [1]; our contribution is demonstrating how targeted verifier feedback mitigates this issue (Section 4.4).
> >
> > **W8 (Experimental details):** Comprehensive setup details are provided in Appendix J (J.2 for model specifications) to accommodate page limits.
> >
> > **W9 (Focus and convincingness):** Our work presents a unified narrative supporting one core claim: effective RLVR for theorem proving requires a specialized verifier. We first validate verifier robustness (Sections 4.1-4.3), then demonstrate its unique impact on RLVR performance (Section 4.4).
> >
> > We hope these clarifications address the reviewer's concerns and demonstrate the rigor of our experimental validation.

---

### Author Response · Authors · 2025-11-29

We sincerely thank all reviewers for their thoughtful and constructive feedback.

Reviewers consistently highlighted the following strengths of our work:

1. **Novelty and Motivation:** The approach of leveraging formal proofs to train informal verifiers is well-motivated, novel, and addresses an important problem in mathematical reasoning (Reviewers 7qVZ, EBzS, Ms64)

2. **Technical Contribution:** The formal-to-informal translation pipeline is clever and represents a first attempt at using formal feedback to bootstrap informal verification ability, which is of significant importance (Reviewers 7qVZ, EBzS, Ms64)

3. **Clarity and Experimental Rigor:** The paper is well-written with rich experiments, theoretical analysis, and helpful case studies demonstrating the verifier's capabilities (Reviewers 7qVZ, EBzS)

4. **Strong Verification Performance:** The trained verifier achieves high verification accuracy (93%) with good calibration and high correlation to human annotators (Reviewer EBzS)

In response to reviewers' concerns, we provide the following additional experimental results:

1. **Frontier Model Comparisons:** We add comparisons with GPT-5 (0.77/0.67) and Gemini-2.5-Pro (0.89/0.73) versus Ours (0.93/0.91) on Natural/Formal Language benchmarks, addressing concerns about incomplete baselines (Reviewers 7qVZ, EBzS).

2. **Generalization Validation:** We provide OPC results (84.1% Pass@1) and comprehensive AIME24 Best-of-N evaluations (0.75 vs. 0.37 baseline at N=32), demonstrating effective generalization to out-of-distribution traditional math reasoning tasks and addressing distribution misalignment concerns (Reviewers 7qVZ, Ms64).

3. **Complete Ablation Studies:** Table 2's Data and Method Ablation sections comprehensively validate each pipeline component's effectiveness (0.82→0.91 accuracy improvement) (Reviewer pLUk).

4. **Additional Metrics:** We supplement existing results with TPR/TNR values and complete Best-of-N curves across all sampling budgets (N=1,2,4,8,16,32) (Reviewers pLUk, EBzS).

---

### Meta-Review · Area_Chair_dtH4 · 2026-01-07

**Summary:**

Here is  a summary of the reviewers' concerns

 - Insufficient baselines or  positioning vs prior art. Missing comparisons to alternative data generation (e.g., human-written corpora + perturbations) and to SOTA generative reward models. Some comparisons focus on DeepSeek-R1 only, which may be too weak (pLUk, EBzS, 7qVZ).
 - End-to-end “enables RLVR” not convincingly demonstrated. Reviewers want stronger, full-scale RLVR training loops with clearer benchmark evaluations and stronger baselines (base model, other RM, ground-truth rewards) beyond preliminary or limited evidence (pLUk, Ms64, 7qVZ).
 - Formal-proof verification usefulness questioned. Since compilers can verify formal proofs, reviewer doubts this is a meaningful target unless transfer benefits are clearly demonstrated (pLUk).
 - Generalization and distribution shift risk. Concern that supervision from formal→informal transformations may not match real informal solver error modes. Plus uncertainty about OOD generalization to trajectories from unseen model families (7qVZ, EBzS).
 - Evaluation integrity of informal test set. Only \approx 100 expert-labeled items. The rest labeled by Gemini, raising the risk that results partly measure agreement with the judge rather than human truth (EBzS).
 - Reporting + reproducibility gaps. Unclear or underspecified metrics (e.g., “correlation” vs full diagnostic stats), unclear definitions of “single-sample” vs Best-of-N, and missing experimental details or protocols in the main text (pLUk, EBzS, Ms64).
 - Ablations not isolating value of components. Reviewer notes minimal deltas between baseline and final pipeline in method ablations, leaving uncertainty about which stages are truly necessary (pLUk).
 - Clarity and scope concerns. Paper tries to do too many things but doesn't manage to be fully convincing for any of them. Writing issues (vague abstract, bulky pseudocode) reduce persuasive power (pLUk, Ms64).

**Reviewer Concerns:**

Here are some of not fully addressed by the rebuttal remaining concerns.

 - Informal “ground truth” still largely LLM-judged (only approx. 100 expert-labeled anchors; rest Gemini-labeled), so conclusions may partly reflect judge-consistency rather than human-truth (EBzS).
 - OOD generalization / distribution-shift not conclusively established, formal to informal supervision may not match real informal solver error modes, and robustness across unseen prover families/trajectories remains uncertain (7qVZ, EBzS).
 - End-to-end RLVR claim still under-supported for its centrality, reviewers wanted stronger benchmark-style validation of the RL-trained prover against multiple baselines (base, other RM, ground-truth reward), not just limited RLVR evidence plus some added tasks (Ms64, pLUk).
 - Baseline breadth + component necessity remain unclear, comparisons to alternative pipelines/SOTA generative RMs and ablations that cleanly identify which pipeline pieces matter may still be insufficient (pLUk, 7qVZ).

**Reviewer Scores:**

In the view of remaining issues the scores would have probably remained unchanged

---

### Decision · Program_Chairs · 2026-01-26

Reject